# Natural variation in *C. elegans* arsenic toxicity is explained by differences in branched chain amino acid metabolism

Stefan Zdraljevic[1,2], Bennett William Fox[3], Christine Strand[4], Oishika Panda[3,5], Francisco J Tenjo[3], Shannon C Brady[1,2], Tim A Crombie[2], John G Doench[4], Frank C Schroeder[3], Erik C Andersen[1,2,6]*

[1]Interdisciplinary Biological Sciences Program, Northwestern University, Evanston, United States; [2]Department of Molecular Biosciences, Northwestern University, Evanston, United States; [3]Boyce Thompson Institute and Department of Chemistry and Chemical Biology, Cornell University, Ithaca, United States; [4]Broad Institute of MIT and Harvard, Cambridge, United States; [5]The Buck Institute for Research on Aging, Novato, United States; [6]Robert H. Lurie Comprehensive Cancer Center of Northwestern University, Northwestern University, Chicago, United States

**Abstract** We find that variation in the *dbt-1* gene underlies natural differences in *Caenorhabditis elegans* responses to the toxin arsenic. This gene encodes the E2 subunit of the branched-chain α-keto acid dehydrogenase (BCKDH) complex, a core component of branched-chain amino acid (BCAA) metabolism. We causally linked a non-synonymous variant in the conserved lipoyl domain of DBT-1 to differential arsenic responses. Using targeted metabolomics and chemical supplementation, we demonstrate that differences in responses to arsenic are caused by variation in iso-branched chain fatty acids. Additionally, we show that levels of branched chain fatty acids in human cells are perturbed by arsenic treatment. This finding has broad implications for arsenic toxicity and for arsenic-focused chemotherapeutics across human populations. Our study implicates the BCKDH complex and BCAA metabolism in arsenic responses, demonstrating the power of *C. elegans* natural genetic diversity to identify novel mechanisms by which environmental toxins affect organismal physiology.

**Editorial note:** This article has been through an editorial process in which the authors decide how to respond to the issues raised during peer review. The Reviewing Editor's assessment is that all the issues have been addressed (see decision letter).

DOI: https://doi.org/10.7554/eLife.40260.001

*For correspondence:
erik.andersen@northwestern.edu

Competing interests: The authors declare that no competing interests exist.

## Introduction

An estimated 100 million people are currently at risk of chronic exposure to arsenic, a toxic metalloid that can be found in the environment (*Ravenscroft et al., 2009*). The high prevalence of environmental arsenic and the severe toxicity associated with exposure has made it the number one priority for the United States Agency for Toxic Substances and Disease Registry (https://www.atsdr.cdc.gov/SPL/). Inorganic trivalent arsenic As(III) compounds, which include arsenic trioxide ($As_2O_3$), are the most toxic forms of environmental arsenic (*Ratnaike, 2003; Mandal and Suzuki, 2002*). In humans, As(III) is detoxified by consecutive methylation events, forming dimethylarsenite (DMA) (*Khairul et al., 2017; Stýblo et al., 2002*). However, this methylation process also creates the highly toxic monomethylarsenite (MMA) intermediate, so ratios of DMA to MMA determine levels of arsenic toxicity. Both MMA and DMA are produced from As(III) via the arsenic methyltransferase (AS3MT) (*Schlebusch et al., 2015*). Interestingly, individuals from human subpopulations that inhabit

high arsenic environments have higher DMA/MMA ratios than individuals from low-arsenic environments. The elevated DMA/MMA ratio in these individuals is associated with natural differences in the *AS3MT* gene (*Chung et al., 2009*; *Fujihara et al., 2009*; *Gomez-Rubio et al., 2010*), which shows signs of strong positive selection. These results suggest that a more active AS3MT enzyme in these human subpopulations makes more DMA and enables adaptation to elevated environmental arsenic levels (*Schlebusch et al., 2015*). Importantly, population-wide differences in responses to environmental arsenic cannot be explained solely by variation in *AS3MT*, indicating that other genes must impact arsenic toxicity.

Despite its toxicity, arsenic trioxide has been used as a therapeutic agent for hundreds of years. Most recently, it was introduced as a highly effective cancer chemotherapeutic for the treatment of acute promyelocytic leukemia (APL) (*Chen et al., 1997*; *Antman, 2001*; *Murgo, 2001*; *Emi, 2017*). Hematopoietic differentiation and apoptosis in APL patients is blocked at the level of promyelocytes by the Promyelocytic Leukemia/Retinoic Acid Receptor alpha fusion protein caused by a t(15;17) chromosomal translocation (*de Thé et al., 1990*; *Grignani et al., 2000*). Arsenic trioxide has been shown to directly bind a cysteine-rich region of the RING-B box coiled-coil domain of PML-RARα, which causes the degradation of the oncogenic fusion protein (*Zhang et al., 2010*; *Tomita et al., 2013*). The success of arsenic trioxide (Trisenox) has spurred its use in over a hundred clinical trials in the past decade (*Hoonjan et al., 2018*). Despite these successes, individual differences in the responses to arsenic-based treatments, including patient-specific dosing regimens and side effects, limit the full therapeutic benefit of this compound (*Zeidan and Gore, 2014*). Medical practitioners require knowledge of the molecular mechanisms for how arsenic causes toxicity to provide the best individual-based therapeutic benefits.

Studies of the free-living roundworm *Caenorhabditis elegans* have greatly facilitated our understanding of basic cellular processes (*Kniazeva et al., 2004*; *Luz et al., 2017*; *Spracklin et al., 2017*; *Watson et al., 2013*), including a number of studies that show that the effects of arsenic are similar to what is observed in mammalian model systems and humans. These effects include mitochondrial toxicity (*Luz and Meyer, 2016*; *Luz et al., 2016*), the generation of reactive oxygen species (ROS) (*Schmeisser et al., 2013*), genotoxicity (*Wyatt et al., 2017*), genome-wide shifts in chromatin structure (*Large et al., 2016*), reduced lifespan (*Schmeisser et al., 2013*), and the induction of the heat-shock response (*Wang et al., 2017*). However, these studies were all performed in the genetic background of the standard *C. elegans* laboratory strain (N2). To date, 152 *C. elegans* strains have been isolated from various locations around the world (*Andersen et al., 2012*; *Cook et al., 2016*; *Cook et al., 2017*), which contain a largely unexplored pool of genetic diversity much of which could underlie adaptive responses to environmental perturbations (*Zdraljevic and Andersen, 2017*).

We used two quantitative genetic mapping approaches to show that a major source of variation in *C. elegans* responses to arsenic trioxide is caused by natural variation in the *dbt-1* gene, which encodes an essential component of the highly conserved branched-chain α-keto acid dehydrogenase (BCKDH) complex (*Jia et al., 2016*). The BCKDH complex is a core component of branched-chain amino acid (BCAA) catabolism, which has not been previously implicated in arsenic responses. Furthermore, we show that a single missense variant in DBT-1(S78C), located in the highly conserved lipoyl-binding domain, underlies phenotypic variation in response to arsenic. Using targeted and untargeted metabolomics and chemical rescue experiments, we show that differences in wild isolate responses to arsenic trioxide are caused by differential synthesis of mono-methyl branched chain fatty acids (mmBCFA), metabolites with a central role in development (*Kniazeva et al., 2004*). These results demonstrate the power of using the natural genetic diversity across the *C. elegans* species to identify mechanisms by which environmental toxins affect physiology.

## Results

### Natural variation on chromosome II underlies differences in arsenic responses

We quantified arsenic trioxide sensitivity in *C. elegans* using a high-throughput fitness assay that utilizes the COPAS BIOSORT (*Andersen et al., 2015*; *Zdraljevic and Andersen, 2017*). In this assay, three L4 larvae from each strain were sorted into arsenic trioxide or control conditions. After four days of growth, we quantified various attributes of populations that relate to the ability of *C. elegans*

to grow in the presence of arsenic trioxide or control conditions (see Materials and methods). To determine an appropriate concentration of arsenic trioxide for mapping experiments, we performed dose-response experiments on four genetically diverged isolates of *C. elegans*: N2, CB4856, JU775, and DL238 (*Figure 1—figure supplement 1*; *Figure 1—figure supplement 1—source data 1*). To assess arsenic-induced toxicity, we studied four independently measured traits: brood size, animal length, optical density, and fluorescence (see Materials and methods). When compared to control conditions, all four strains produced fewer progeny at all arsenic trioxide concentrations, and the lowest concentration at which we observed a significant reduction in brood size for all strains was 1 mM (*Figure 1—figure supplement 1A*). We used statistical summaries of measurements of individual animals as replicated identical genotypes to estimate broad-sense heritability ($H^2$) across the four strains, but this analysis might not represent the effects of arsenic on individuals within natural populations. For the brood size trait in 1 mM arsenic trioxide, we calculated $H^2$ to be 0.65 (*Figure 1—figure supplement 3—source data 1*) and the strain effect to be 0.48 (partial omega squared, $\omega_p^2$, *Figure 1—figure supplement 3—source data 2*), indicating that this trait has a large genetic component and a large strain effect. In addition to brood size effects, we observed that the progeny of animals exposed to arsenic trioxide were shorter in length than the progeny of animals grown in control conditions (*Figure 1—figure supplement 1B*), which indicates an arsenic-induced developmental delay (animal length (mean.TOF) $H^2$ = 0.13; *Figure 1—figure supplement 3—source data 1* and $\omega_p^2$ = 0.09; *Figure 1—figure supplement 3—source data 2*). As *C. elegans* develop, the animals increase in optical density. Therefore, it is not surprising that we found an arsenic-induced decrease in optical density of progeny populations, which is further support for an arsenic-induced developmental delay (*Figure 1—figure supplement 1C*; $H^2$ = 0.19; *Figure 1—figure supplement 3—source data 1* and $\omega_p^2$ = 0.07; *Figure 1—figure supplement 3—source data 2*). We also observed an arsenic-induced effect on yellow autofluorescence (*Figure 1—figure supplement 1D*; $H^2$ = 0.50; *Figure 1—figure supplement 3—source data 1* and $\omega_p^2$ = 0.21; *Figure 1—figure supplement 3—source data 2*). Overall, the CB4856 strain was less affected by arsenic – that strain produced approximately 16% more offspring that were on average 20% larger than the other three strains when treated with 1 mM arsenic trioxide. These results suggest that the CB4856 strain was more resistant to arsenic trioxide than the other three strains. In addition to the BIOSORT-quantified traits, we generated a synthetic principal component (PC) trait using the four quantified traits described above (see Materials and methods, *Figure 1—figure supplement 1—source data 1*, *Figure 1—figure supplement 2—source data 1* and *Figure 1—source data 1*). For 1 mM arsenic trioxide, we estimated the broad-sense heritability ($H^2$) of the first principal component to be 0.37 (*Figure 1—figure supplement 3—source data 1*) with an effect size of 0.001 ($\omega_p^2$) (*Figure 1—figure supplement 3—source data 2*). The first principal component explained a large fraction (0.75) of the total phenotypic variance within the experiment, which is likely because the four input traits correlate with arsenic-induced toxicity (*Figure 1—figure supplement 2A*; *Figure 1—figure supplement 1—source data 3*). We noted that the first principal component (PC1) was most strongly influenced by the optical density trait, as indicated by the loadings (*Figure 1—figure supplement 2B*; *Figure 1—figure supplement 3—source data 2* and *Figure 1—source data 1*), suggesting that PC1 is a biologically relevant trait (*Figure 1—figure supplement 1E*). Furthermore, because we observe a large range of effect sizes and broad-sense heritability estimates across measured traits (*Figure 1—figure supplement 3*), we focused our analyses on the PC1 trait derived from the four BIOSORT-quantified described above for all subsequent experiments (Materials and methods).

The increased arsenic trioxide resistance of CB4856 compared to N2 motivated us to perform linkage mapping experiments with a panel of recombinant inbred advanced intercross lines (RIAILs) that were previously constructed through ten generations of intercrossing between an N2 derivative (QX1430) and CB4856 (*Andersen et al., 2015*). To capture arsenic trioxide-induced phenotypic differences, we exposed a panel of 252 RIAILs to 1 mM arsenic trioxide and corrected for growth differences among RIAILs in control conditions and assay-to-assay variability using linear regression (*Figure 1—source data 2*; see Materials and methods). We performed linkage mapping on processed traits and the eigenvector-transformed traits (principal components or PCs) obtained from PCA that explained 90% of the variance in the processed trait set (Materials and methods). The rationale of this approach was to minimize trait fluctuations that could be caused by only measuring

the phenotypes of one replicate per RIAIL strain, and PC1 captured overall arsenic-induced toxicity. In agreement with our observations from the dose-response experiment, we found that PC1 captures 69.5% of the total measured trait variance and is strongly influenced by animal length traits (*Figure 1—figure supplement 4*; *Figure 1—figure supplement 11—source data 1* and *2*). Linkage mapping analysis of the PC1 trait revealed that arsenic trioxide-induced phenotypic variation is significantly associated with genetic variation on the center of chromosome II (*Figure 1—figure supplement 4*; *Figure 1—source data 3*). An additional quantitative trait locus (QTL) was significantly associated with variation in arsenic responses on chromosome X (*Figure 1—figure supplement 4*). Consistent with the loadings of PC1, we determined that PC1 is highly correlated with the both brood size and animal length traits (*Figure 1B*), suggesting that PC1 captures RIAIL variation in these traits. To further support this relationship to interpretable biological significance, we found that the four traits used as input for PCA all map to the same region on the center of chromosome II (*Figure 1—figure supplement 5*; *Figure 1—source data 3*). The QTL on the center of chromosome II explains 35.4% of the total RIAIL phenotypic variation for the PC1 trait, which accounts for 63.4% of the total phenotypic variation that can be explained by genetic factors ($H^2 = 0.56$) (*Figure 1—figure supplement 6*; *Figure 1—figure supplement 6—source data 1*). Taken together, the two QTL identified by mapping the PC1 trait account for 40% of the total RIAIL variation, corresponding to 71.6% of the total phenotypic variation that can be explained by genetic factors. However, we did not account for errors in genomic heritability estimates. In addition to the two QTL that explain variation of the PC1 trait, we identified a QTL on chromosome I for the brood size and optical density traits, and a QTL on chromosome V that explained variation in animal length and optical density upon arsenic exposure (*Figure 1—figure supplement 7*; *Figure 1—source data 3*). The PC1 QTL confidence interval spans from 7.04 to 8.87 Mb on chromosome II. This QTL overlaps with the brood size (6.18-9.31 Mb) and animal length (6.92-8.70 Mb) QTL confidence intervals and is identical to the optical density and fluorescence QTL (*Figure 1—figure supplement 5*; *Figure 1—source data 3*). However, each of these QTL confidence intervals span genomic regions greater than 1.5 megabases and contain hundreds of genes that vary between the N2 and CB4856 strains.

Next, we constructed near-isogenic lines (NILs) to isolate and narrow the chromosome II QTL in a controlled genetic background. We introgressed genomic regions from the CB4856 strain on the left and right halves of the confidence interval into the N2 genetic background. In the presence of arsenic trioxide, both of these NILs recapitulated the parental CB4856 PC1 phenotype (*Figure 1C*; *Figure 1—source data 4*) and had similar trait values for the four traits used as inputs into the PCA (*Figure 1—figure supplement 8*; *Figure 1—source data 4*). Furthermore, we showed that similar to the RIAIL phenotypes, the measured traits were correlated (*Figure 1—figure supplement 9A*; *Figure 1—figure supplement 9—source data 1*) and contributed similarly to the PC1 trait (*Figure 1—figure supplement 9B*; *Figure 1—figure supplement 9—source data 2*). Furthermore, the PC1 trait was highly correlated with the four input traits (*Figure 1—figure supplement 10*). The phenotypic similarity of these NILs to the CB4856 parental strain suggested that the two NILs might share an introgressed region of the CB4856 genome. To identify this shared introgressed region, we performed low-coverage whole-genome sequencing of the NIL strains and defined the left and right bounds of the CB4856 genomic introgression to be from 5.75 to 8.02 Mb and 7.83 to 9.66 Mb in the left and right NILs, respectively (*Figure 1—source data 5*). The left and right NILs recapitulate 70.6% and 81.9% of the effect size difference between N2 and CB4856 as measured by Cohen's F, respectively (*Cohen, 2013*), which exceeds our observations the linkage mapping results where the QTL on chromosome II explained 63.4% of the total phenotypic variation in the RIAIL population. This discrepancy was observed likely because the NILs are a more homogenous genetic background, and the experiment was performed at higher replication than the linkage mapping. We observed similar levels of phenotypic recapitulation for the four traits used as inputs for the PCA (brood size: 57.8% and 87.4%, animal length: 98.5% and 100%; 69.5% and 68.1%; fluorescence: 58.5% and 64.2% for the left and right NILs). Taken together, these results suggested that genetic differences between N2 and CB4856 within 7.83 to 8.02 Mb on chromosome II conferred resistance to arsenic trioxide.

In parallel to the linkage-mapping approach described above, we performed a genome-wide association (GWA) mapping experiment by quantifying the responses to arsenic trioxide for 86 wild *C. elegans* strains (*Figure 2—figure supplement 1—source data 1*) (*Andersen et al., 2012*). Consistent with previous experiments, the PC1 trait was influenced less by the brood size trait, as indicated

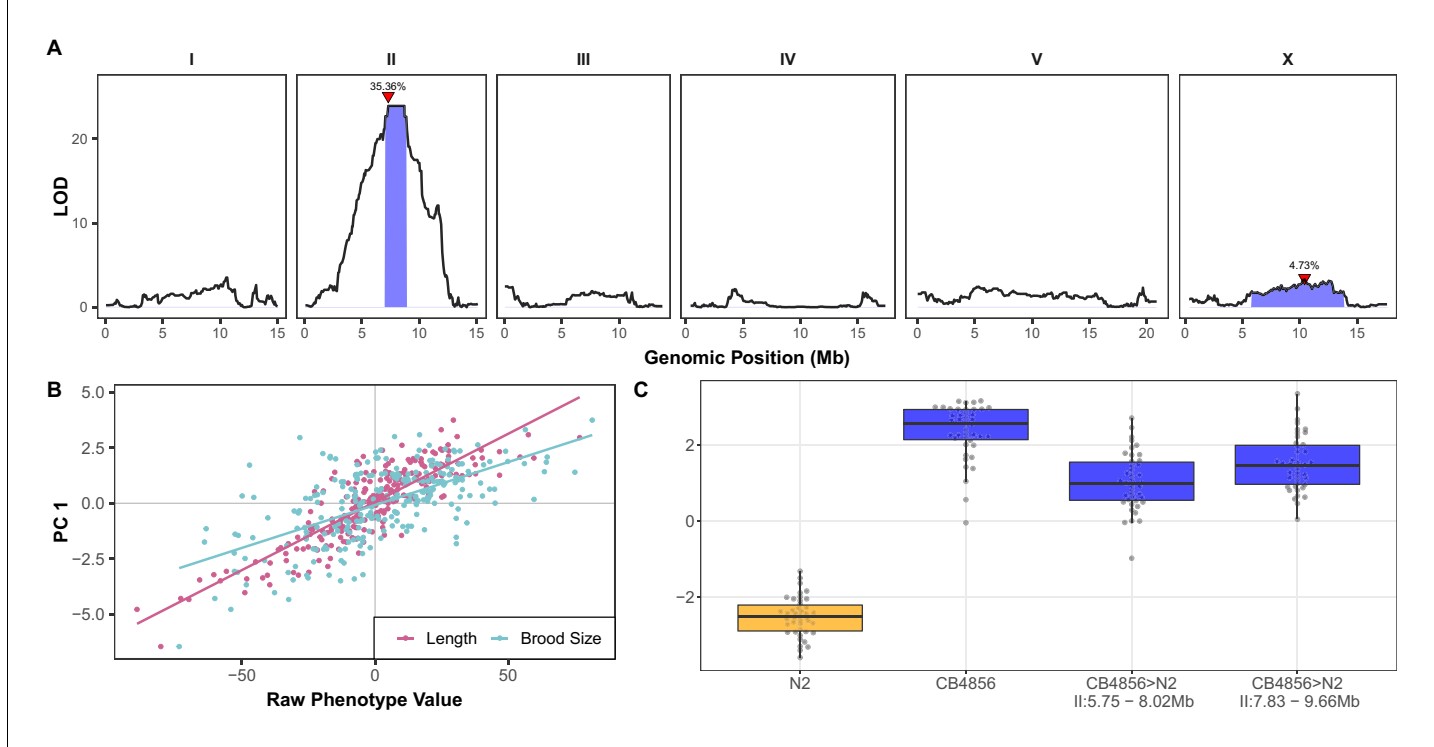

**Figure 1.** A large-effect QTL on the center of chromosome II explains differences in arsenic trioxide response between N2 and CB4856. (**A**) Linkage mapping plots for the first principal component trait in the presence of 1000 μM arsenic trioxide is shown. The significance values (logarithm of odds, LOD, ratio) for 1454 markers between the N2 and CB4856 strains are on the y-axis, and the genomic position (Mb) separated by chromosome is plotted on the x-axis. The associated 1.5 LOD-drop confidence intervals are represented by blue boxes. The phenotypic variance explained by each QTL is shown above the peak QTL marker, which is marked by red triangles. (**B**) The correlation between brood size (blue; $r^2 = 0.38$, p-value=1.65E-27) or animal length (pink; $r^2 = 0.74$, p-value=3.16E-74) with the first principal component trait. Each dot represents an individual RIAIL's phenotype, with the animal length and brood size phenotype values on the x-axis and the first principal component phenotype on the y-axis. (**C**) Tukey box plots of near-isogenic line (NIL) phenotype values for the first principal component trait in the presence of 1000 μM arsenic trioxide is shown. NIL genotypes are indicated below the plot as genomic ranges. The N2 trait is significantly different than the CB4856 and NIL traits (Tukey HSD p-value<1E-5).

DOI: https://doi.org/10.7554/eLife.40260.002

The following source data and figure supplements are available for figure 1:

**Source data 1.** Arsenic dose-response loadings of principal components (PCs) for the PCs that explain up to 90% of the total variance in the trait data.
DOI: https://doi.org/10.7554/eLife.40260.025

**Source data 2.** RIAIL phenotype data used for linkage mapping.
DOI: https://doi.org/10.7554/eLife.40260.026

**Source data 3.** Results from linkage mapping experiment.
DOI: https://doi.org/10.7554/eLife.40260.027

**Source data 4.** Phenotypes of NILs and CRISPR allele replacement strains in the presence of 1000 μM arsenic trioxide after correcting for strain differences in control conditions.
DOI: https://doi.org/10.7554/eLife.40260.028

**Source data 5.** NIL genotypes generated from whole-genome sequencing.
DOI: https://doi.org/10.7554/eLife.40260.029

**Figure supplement 1.** Arsenic trioxide dose response of four diverged *C. elegans* strains.
DOI: https://doi.org/10.7554/eLife.40260.003

**Figure supplement 1—source data 1.** Arsenic dose-response trait data for four strains in 0, 250, 500, 1000, or 2000 μM arsenic, labeled as water, arsenic1, arsenic2, arsenic3, arsenic4 in the Condition column.
DOI: https://doi.org/10.7554/eLife.40260.004

**Figure supplement 1—source data 2.** Arsenic dose-response principal component (PC) eigenvectors for the PCs that explain up to 90% of the total variance in the data set.
DOI: https://doi.org/10.7554/eLife.40260.005

**Figure supplement 1—source data 3.** Arsenic dose-response trait correlations where each row corresponds to the Pearson correlation coefficient for two traits.

*Figure 1 continued on next page*

*Figure 1 continued*

DOI: https://doi.org/10.7554/eLife.40260.006

**Figure supplement 2.** Trait correlations and principal component loadings of arsenic trioxide dose response.

DOI: https://doi.org/10.7554/eLife.40260.007

**Figure supplement 2—source data 1.** Arsenic dose-response loadings of principal components (PCs) for the PCs that explain up to 90% of the total variance in the trait data.

DOI: https://doi.org/10.7554/eLife.40260.008

**Figure supplement 3.** Effect size and broad-sense heritability estimates for the arsenic trioxide dose response.

DOI: https://doi.org/10.7554/eLife.40260.009

**Figure supplement 3—source data 1.** Broad-sense heritability estimates from arsenic dose response data.

DOI: https://doi.org/10.7554/eLife.40260.010

**Figure supplement 3—source data 2.** Arsenic dose-response estimates of effect sizes estimated by fitting the linear model: trait value ~strain.

DOI: https://doi.org/10.7554/eLife.40260.011

**Figure supplement 4.** RIAIL phenotypes from the linkage mapping experiment.

DOI: https://doi.org/10.7554/eLife.40260.012

**Figure supplement 5.** Linkage mapping results for brood size, animal length, and the first principal component.

DOI: https://doi.org/10.7554/eLife.40260.013

**Figure supplement 6.** Genomic-heritability estimates of linkage mapping traits.

DOI: https://doi.org/10.7554/eLife.40260.014

**Figure supplement 6—source data 1.** Genomic heritability estimates from RIAIL phenotype data.

DOI: https://doi.org/10.7554/eLife.40260.015

**Figure supplement 7.** Linkage mapping QTL summary.

DOI: https://doi.org/10.7554/eLife.40260.016

**Figure supplement 8.** NIL recapitulation of chromosome II QTL.

DOI: https://doi.org/10.7554/eLife.40260.017

**Figure supplement 9.** Trait correlations and principal component loadings of NIL and allele-replacement recapitulation experiment.

DOI: https://doi.org/10.7554/eLife.40260.018

**Figure supplement 9—source data 1.** NIL and CRISPR allele replacement trait correlations.

DOI: https://doi.org/10.7554/eLife.40260.019

**Figure supplement 9—source data 2.** NIL and CRISPR allele replacement trait loadings of principal components (PCs) for the PCs that explain up to 90% of the total variance in the trait data.

DOI: https://doi.org/10.7554/eLife.40260.020

**Figure supplement 10.** Brood size and animal length are correlated with the first principal component for the NIL recapitulation experiment.

DOI: https://doi.org/10.7554/eLife.40260.021

**Figure supplement 11.** Trait correlations and principal component loadings of linkage mapping experiment.

DOI: https://doi.org/10.7554/eLife.40260.022

**Figure supplement 11—source data 1.** RIAIL trait correlations where each row corresponds to the Pearson correlation coefficient for two traits.

DOI: https://doi.org/10.7554/eLife.40260.023

**Figure supplement 11—source data 2.** RIAIL loadings of principal components (PCs) for the PCs that explain up to 90% of the total variance in the trait data.

DOI: https://doi.org/10.7554/eLife.40260.024

---

by the loadings (*Figure 2—figure supplement 1*; *Figure 2—source data 1* and *Figure 5—figure supplement 1—source data 1*). In agreement with the results from the linkage mapping approach, PC1 differences among the wild isolates mapped to a QTL on the center of chromosome II that spans from 7.6 Mb to 8.21 Mb (*Figure 2A*; *Figure 2—figure supplement 3—source data 1* and *Figure 2—figure supplement 5—source data 1*). However, we noted that the brood size trait did not map to a significant QTL with the GWA mapping approach, which is most likely due to the lower statistical power of this approach. Interestingly, the genomic estimates of broad- and narrow-sense heritability ($H^2$; $h^2$) were low for all of the wild isolates measured and principal component traits (*Figure 2—figure supplement 2*; *Figure 2—source data 2*), which could be because the center of chromosome II has not experienced the chromosome-scale selective sweeps (*Andersen et al., 2012*) that contribute to much of the population structure within the species. The marker found to be most correlated with the PC1 trait from GWA mapping (II:7,931,252), explains 84.6% of the total heritable phenotypic variation. In addition to the PC1 trait, three of the four measured traits also mapped to significant QTL on the center of chromosome II. (*Figure 2—figure supplement 1*; *Figure 2—figure*

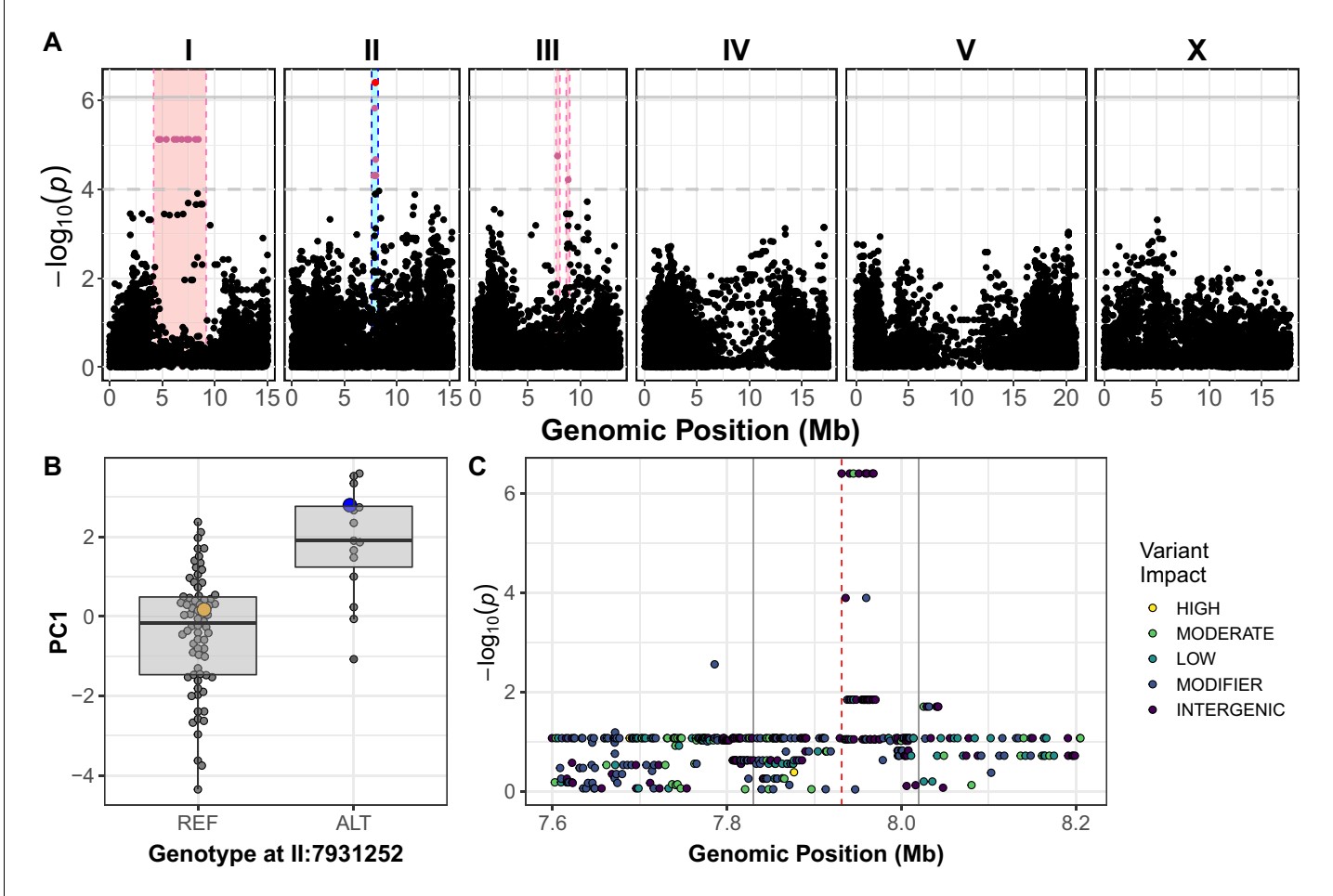

**Figure 2.** Variation in *C. elegans* wild isolates responses to arsenic trioxide maps to the center of chromosome II. (**A**) A manhattan plot for the first principal component in the presence of 1000 µM arsenic trioxide is shown. Each dot represents an SNV that is present in at least 5% of the assayed wild population. The genomic position in Mb, separated by chromosome, is plotted on the x-axis and the *-log₁₀(p)* for each SNV is plotted on the y-axis. SNVs are colored red if they pass the genome-wide Bonferroni-corrected significance (BF) threshold, which is denoted by the gray horizontal line. SNVs are colored pink if they pass the genome-wide eigen-decomposition significance (ED) threshold, which is denoted by the dotted gray horizontal line. The genomic region of interests surrounding the QTL that pass the BF and ED thresholds are represented by cyan and pink rectangles, respectively. (**B**) Tukey box plots of phenotypes used for association mapping in (**A**) are shown. Each dot corresponds to the phenotype of an individual strain, which is plotted on the y-axis. Strains are grouped by their genotype at the peak QTL position (red SNV from panel A, ChrII:7,931,252), where REF corresponds to the allele from the reference N2 strain. The N2 (orange) and CB4856 (blue) strains are highlighted. (**C**) Fine mapping of the chromosome II region of interest (cyan region from panel A, 7.60–8.21 Mb) is shown. Each dot represents an SNV present in the CB4856 strain. The association between the SNV and first principal component is shown on the y-axis and the genomic position of the SNV is shown on the x-axis. Dots are colored by their SnpEff predicted effect.

DOI: https://doi.org/10.7554/eLife.40260.030

The following source data and figure supplements are available for figure 2:

**Source data 1.** Fine-mapping results for PC1.
DOI: https://doi.org/10.7554/eLife.40260.040
**Source data 2.** All wild-isolate traits used for genome-wide association mapping.
DOI: https://doi.org/10.7554/eLife.40260.041
**Source data 3.** Genotype matrix used for genome-wide mapping.
DOI: https://doi.org/10.7554/eLife.40260.044
**Source data 4.** GWA mapping results for PC1.
DOI: https://doi.org/10.7554/eLife.40260.045
**Figure supplement 1.** GWA mapping QTL summary.
DOI: https://doi.org/10.7554/eLife.40260.031
**Figure supplement 1—source data 1.** All QTL identified by GWA mapping (Used to generate *Figure 2—figure supplement 1*).

*Figure 2 continued on next page*

*Figure 2 continued*

DOI: https://doi.org/10.7554/eLife.40260.032

**Figure supplement 2.** Fine-mapping of the chromosome II QTL identified by GWA mapping.

DOI: https://doi.org/10.7554/eLife.40260.033

**Figure supplement 3.** Tajima's *D* across the arsenic trioxide QTL confidence interval.

DOI: https://doi.org/10.7554/eLife.40260.034

**Figure supplement 3—source data 1.** Isolation locations of strains used in GWA mapping.

DOI: https://doi.org/10.7554/eLife.40260.035

**Figure supplement 4.** The worldwide distribution of the DBT-1(C78S) allele.

DOI: https://doi.org/10.7554/eLife.40260.036

**Figure supplement 5.** Genomic-heritability estimates of GWA mapping traits.

DOI: https://doi.org/10.7554/eLife.40260.037

**Figure supplement 5—source data 1.** Genomic heritability estimates of wild isolate traits.

DOI: https://doi.org/10.7554/eLife.40260.038

**Figure supplement 6.** Trait correlations and principal component loadings of GWA mapping experiment.

DOI: https://doi.org/10.7554/eLife.40260.039

**Figure supplement 6—source data 1.** Wild isolate trait loadings of principal components (PCs) for the PCs that explain up to 90% of the total variance in the trait data.

DOI: https://doi.org/10.7554/eLife.40260.042

**Figure supplement 6—source data 2.** Wild isolate trait correlations.

DOI: https://doi.org/10.7554/eLife.40260.043

*supplement 1—source data 1*). Notably, the CB4856 strain, which was one of the parents used to construct the RIAIL panel used for linkage mapping, had the non-reference genotype at the marker most correlated with PC1 (*Figure 2B*), suggesting that the same genetic variant(s) might be contributing to the differential arsenic trioxide response between the RIAIL and wild isolate populations.

To fine map the PC1 QTL, we focused on variants from the *C. elegans* whole-genome variation dataset (*Cook et al., 2016*) that are shared among at least 5% of the 86 wild isolates exposed to arsenic trioxide. Under the assumption that the linkage and GWA mapping QTL are caused by the same genetic variation, we only considered variants present in the CB4856 strain. Eight markers within the QTL region are in complete linkage disequilibrium with each other and are most correlated with the PC1 trait (*Figure 2—figure supplement 2*; *Figure 2—source data 1*). Only one of these markers is located within an annotated gene (*dbt-1*) and is predicted to encode a cysteine-to-serine variant at position 78 (C78S). Although it is possible that the causal variant underlying differential arsenic trioxide response in the *C. elegans* population is an intergenic variant, we focused on the DBT-1(C78S) variant as a candidate to test for an effect on arsenic response.

## A cysteine-to-serine variant in DBT-1 contributes to arsenic response variation

The *C. elegans dbt-1* gene encodes the E2 component of the branched-chain α-keto acid dehydrogenase complex (BCKDH) (*Jia et al., 2016*). The BCKDH complex is a core component of branched-chain amino acid (BCAA) catabolism and catalyzes the irreversible oxidative decarboxylation of amino-acid-derived branched-chain α-ketoacids (*Adeva-Andany et al., 2017*). The BCKDH complex belongs to a family of α-ketoacid dehydrogenases that include pyruvate dehydrogenase (PDH) and α-ketoglutarate dehydrogenase (KGDH) (*Bergquist et al., 2009*). All three of these large enzymatic complexes include a central E2 component that is lipoylated at one critical lysine residue (two residues in PDH). The function of these enzymatic complexes depends on the lipoylation of these lysine residues (*Bergquist et al., 2009*; *Reed and Hackert, 1990*). In *C. elegans*, the putative lipoylated lysine residue is located at amino acid position 71 of DBT-1, which is in close proximity to the C78S residue that we found to be highly correlated with arsenic trioxide resistance.

To confirm that the C78S variant in DBT-1 contributes to differential arsenic trioxide responses, we used CRISPR-Cas9-mediated genome editing to generate allele-replacement strains by changing the C78 residue in the N2 strain to a serine and the S78 residue in the CB4856 strain to a cysteine. When treated with arsenic trioxide, the N2 DBT-1(S78) allele-replacement strain recapitulated 56.4% of the phenotypic difference between the CB4856 and N2 strains as measured with the first principal

component (Cohen's F) (*Cohen, 2013*) (*Figure 3*; *Figure 1—source data 4*). Similarly, the CB4856 DBT-1(C78) allele-replacement strain recapitulated 64.8% of the total phenotypic difference between the two parental strains. The degree to which the allele-replacement strains recapitulated the difference in the PC1 trait between the N2 and CB4856 strains matched our observations from the linkage mapping experiment, where the chromosome II QTL explained 63.4% of the total phenotypic variation in the RIAIL population. This result suggested that the majority of heritable variation in arsenic trioxide response was explained by the DBT-1(C78S) allele. We obtained similar results for the BIOSORT-quantified traits (*Figure 3—figure supplement 1*; *Figure 1—source data 4*), suggesting that overall animal physiology is affected by arsenic exposure (*Figure 1—figure supplement 9*; *Figure 3—figure supplement 2*; *Figure 1—figure supplement 11—source data 2* and *Figure 1— source data 3*). However, when considering brood size, the N2 DBT-1(C78S) allele-replacement strain produced an intermediate number of progeny in the presence of arsenic trioxide relative to the parental N2 and CB4856 strains. And the CB4856 DBT-1(S78C) allele-replacement strain produced fewer offspring than both parental strains (*Figure 3—figure supplement 1*; *Figure 1—source data 4*). These results suggested that additional genetic variants between the N2 and CB4856 strains might interact with the DBT-1(C78S) allele to affect different aspects of physiology. Nevertheless, these results functionally validated that the DBT-1 C78S variant underlies differences in physiological responses to arsenic trioxide.

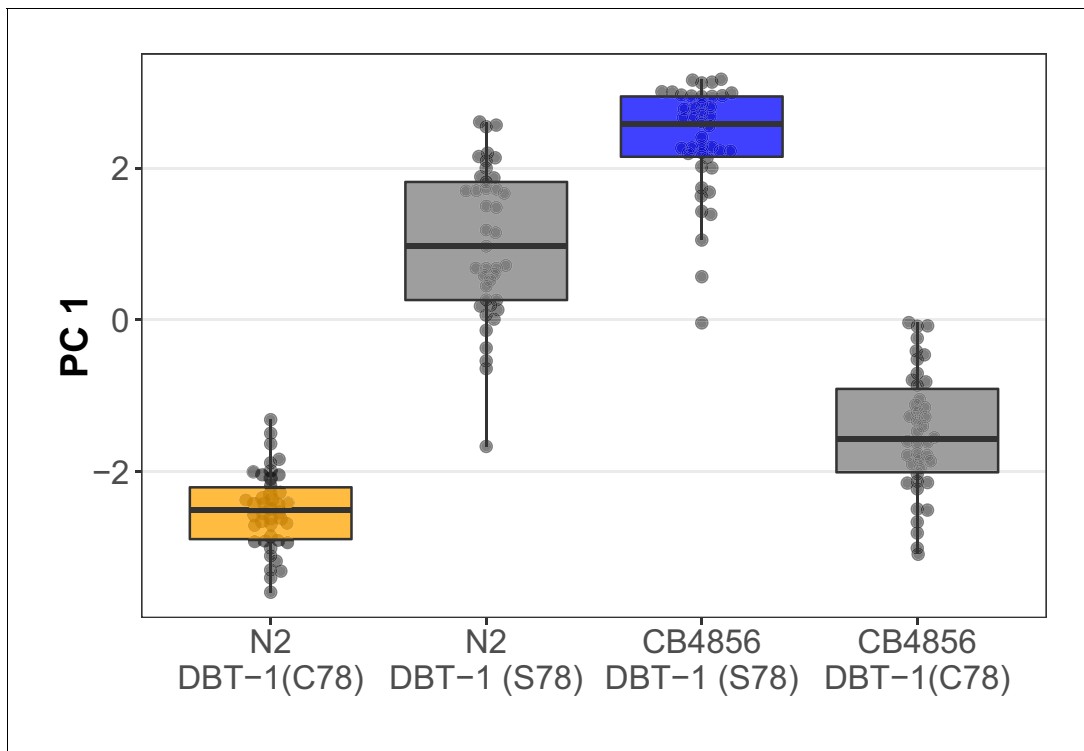

**Figure 3.** The DBT-1(C78S) variant contributes to arsenic trioxide responses. Tukey box plots of the first principal component generated by PCA on allele-replacement strainphenotypes measured by the COPAS BIOSORT 1000 μM arsenic trioxide exposure are shown (N2,orange; CB4856, blue; allele replacement strains, gray). Labels correspond to the genetic backgroundand the corresponding residue at position 78 of DBT-1 (C for cysteine, S for serine). All pair-wise comparisons are significantly different (Tukey HSD, p-value < 1E-7).
DOI: https://doi.org/10.7554/eLife.40260.046

The following figure supplements are available for figure 3:

**Figure supplement 1.** The DBT-1 C78S variant underlies arsenic trioxide sensitivity in *C. elegans*.
DOI: https://doi.org/10.7554/eLife.40260.047

**Figure supplement 2.** Brood size and animal length are correlated with the first principal component for the allele- replacement recapitulation experiment.
DOI: https://doi.org/10.7554/eLife.40260.048

## Arsenic trioxide inhibits the DBT-1 C78 allele

Mono-methyl branched chain fatty acids (mmBCFA) are an important class of molecules that are produced via BCAA catabolism (*Kniazeva et al., 2004*; *Jia et al., 2016*; *Kniazeva et al., 2008*; *Baugh, 2013*). The production of mmBCFA requires the BCKDH, fatty acid synthase (FASN-1), acetyl-CoA carboxylase (POD-2), fatty acyl elongases (ELO-5/6), β-ketoacyl dehydratase (LET-767), and acyl CoA synthetase (ACS-1) (*Kniazeva et al., 2004*; *Jia et al., 2016*; *Kniazeva et al., 2008*; *Watts and Ristow, 2017*; *Entchev et al., 2008*; *Zhu et al., 2013*). Strains that lack functional *elo-5*, *elo-6*, or *dbt-1* produce less C15ISO and C17ISO mmBCFAs, arrest at the L1 larval stage, and can be rescued by supplementing the growth media with C15ISO or C17ISO (*Kniazeva et al., 2004*; *Jia et al., 2016*; *Kniazeva et al., 2008*) (*Figure 4A*).

Because DBT-1 is involved in BCAA catabolism, we hypothesized that the DBT-1(C78S)-dependent difference in progeny length between the N2 and CB4856 strains after arsenic trioxide treatment might be caused by differential larval arrest through depletion of downstream mmBCFAs. To test this hypothesis, we quantified the abundance of the monomethyl-branched (ISO) and straight-chain (SC) forms of C15 and C17 in the N2, CB4856, and allele-replacement genetic backgrounds. We measured the metabolite levels in staged L1 animals and normalized the detected amounts of C15ISO and C17ISO relative to the abundances of C15SC and C17SC, respectively, to mitigate the confounding effects of differences in developmental rates that could be caused by genetic background differences after arsenic trioxide exposure. Generally, the ratios of C15ISO/C15SC and C17ISO/C17SC were reduced in arsenic-treated animals relative to controls (*Figure 4B*; *Figure 4— source datas 1–3*).

However, arsenic trioxide treatment had a 7.6-fold stronger effect on the C15ISO/C15SC ratio in N2, which naturally has the C78 allele, than on the N2 DBT-1(S78) allele replacement strain. This difference suggests that the DBT-1(C78) allele is more strongly inhibited by arsenic trioxide (0.04 to 0.004, Tukey HSD p-value=0.0358, n = 6). Similarly, we observed a 6.6-fold arsenic-induced reduction in the C17ISO/C17SC ratio when comparing the N2 DBT-1(C78) and N2 DBT-1(S78) strains (Tukey HSD p-value=0.003747, n = 6). When comparing the CB4856 DBT-1(S78) and CB4856 DBT-1 (C78) strains, we observed a 2.8-fold lower C15ISO/C15SC ratio (Tukey HSD p-value=0.0427733, n = 3) and 1.5-fold lower C17ISO/C17SC ratio (Tukey HSD p-value=0.164721, n = 3) in the in the CB4856 DBT-1(C78) strain. We noted that the C17ISO/straight-chain ratio difference was not significantly different between the two CB4856 genetic background strains. However, we observed a significant arsenic-induced decrease in raw C17ISO production in the CB4856 DBT-1(C78) strain (Tukey HSD p-value=0.029) and no significant difference in the CB4856 DBT-1(S78) strain (Tukey HSD p-value=0.1) (*Figure 4—figure supplement 1*). Importantly, these DBT-1(C78S) allele-specific reductions in ISO/straight-chain ratios were not caused by arsenic-induced differences in straight-chain fatty acids (*Figure 4—figure supplement 2*). These results explained the majority of the physiological differences between the N2 and CB4856 strains in the presence of arsenic trioxide (*Figure 3*) and suggested that the DBT-1(C78) allele was inhibited by arsenic trioxide more strongly than DBT-1 (S78). Taken together, the differential reduction in branched-chain fatty acids likely underlies the majority of physiological differences between the sensitive and resistant *C. elegans* strains.

In addition to arsenic-induced differences in branched chain fatty acid production, we observed significant differences in branched/straight-chain ratios between the parental and allele replacement strains when L1 larval animals were grown in control conditions (*Figure 4—figure supplement 3*; *Figure 4—source datas 1–3*). Strains with the DBT-1(C78) had higher ISO/SC ratios relative to strains with the DBT-1(S78) for the C17 (CB4856 DBT-1(C78): Tukey HSD p-value=0.0342525, n = 3; N2 DBT-1(C78): Tukey HSD p-value=0.0342525, n = 6) and C15 ratios (CB4856: Tukey HSD p-value=0.0168749, n = 3; N2: Tukey HSD p-value=0.1239674, n = 6) (*Figure 4—figure supplement 3*; *Figure 4—source datas 1–3*). We noted that the C15ISO/straight-chain ratio was not significantly different when comparing the N2 and N2 allele replacement strain, but the direction of effect matched our other observations, and we saw significant differences in C15ISO levels (N2-C15ISO DBT-1(C78): Tukey HSD p-value=0.0265059, n = 6, *Figure 4—figure supplement 4*; *Figure 4— source datas 1–3*). Importantly, the DBT-1 allele-specific differences in the fatty acid ratio and ISO measurements were not caused by differences in straight-chain fatty acids (*Figure 4—figure supplement 4*). However, we did not observe the same effect of the DBT-1(C78S) allele at the young adult life stage (*Figure 4—figure supplement 5*; *Figure 4—figure supplement 4—source data 1*). Taken

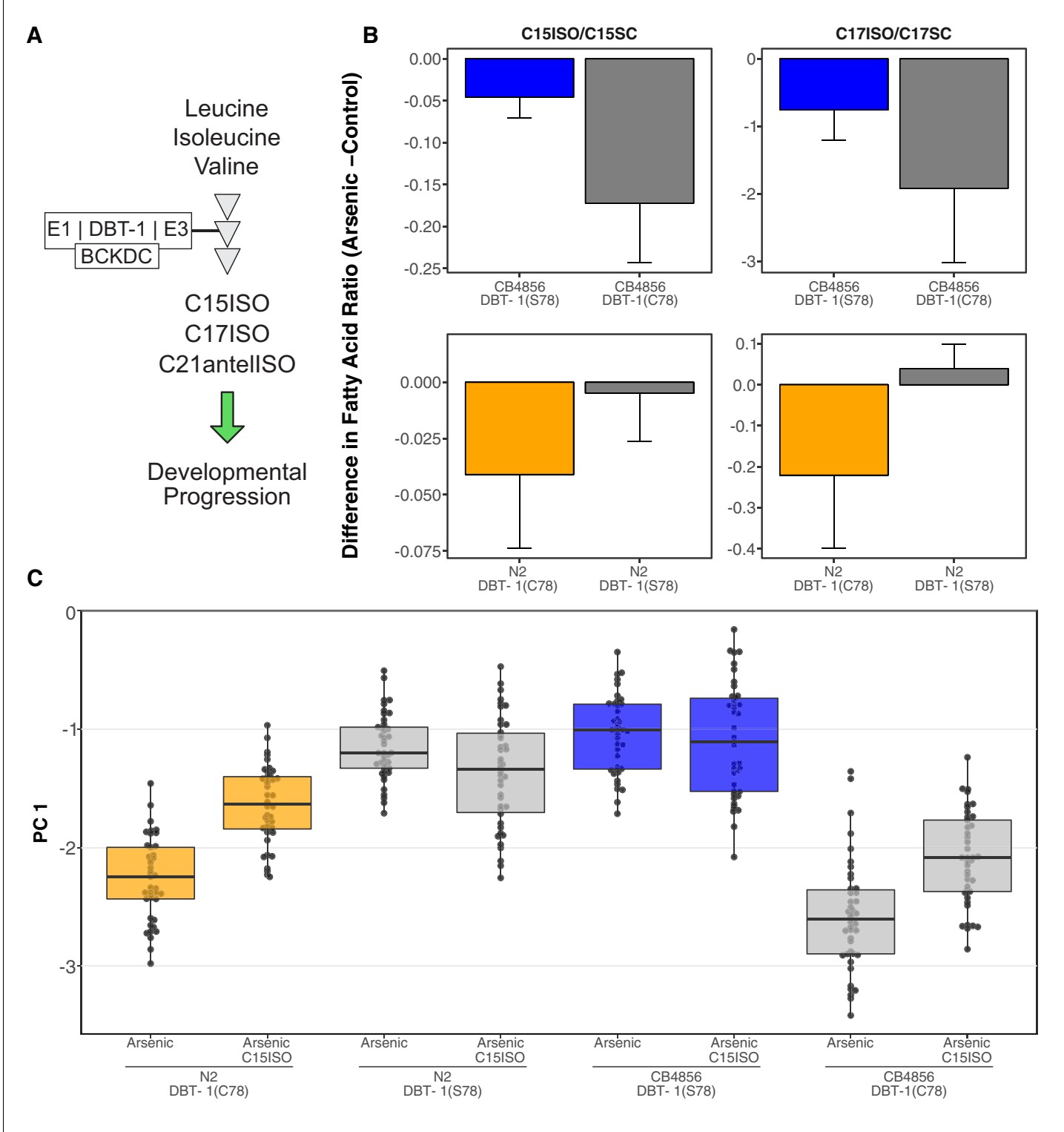

**Figure 4.** Differential production of mmBCFA underlies DBT-1(C78)-mediated sensitivity to arsenic trioxide. (A) A simplified model of BCAA catabolism in *C. elegans*. The BCKDH complex, which consists of DBT-1, catalyzes the irreversible oxidative decarboxylation of branched-chain ketoacids. The products of thesebreakdown can then serve as building blocks for the mmBCFA that are required for developmental progression. (B) The difference in the C15ISO/C15SC (left panel) or C17ISO/C17SC (right panel) ratios between 100 μM arsenic trioxide and control conditions is plotted on the y-axis for three independent replicates of the CB4856 and CB4856 allele replacement strains and six independent replicates of the N2and N2 allele replacement strains. The difference between the C15 ratio for the CB4856-CB4856 allele replacement comparison is significant (Tukey HSD p-value = 0.0427733), but

*Figure 4 continued on next page*

*Figure 4 continued*

the difference between the C17 ratios for these two strains is not (Tukey HSD p-value = 0.164721). The difference between the C15 and C17 ratios for the N2-N2 allele replacement comparisons are both significant (C15: Tukey HSD p-value = 0.0358; C17: Tukey HSD p-value = 0.003747). (**C**) Tukey box plots median animal length after arsenic trioxide or arsenic trioxide and 0.64 μM C15ISO exposure are shown (N2, orange; CB4856, blue; allele replacement strains, gray). Labels correspond to the genetic background and the corresponding residue at position 78 of DBT-1 (C for cysteine, S for serine). Every pair-wise strain comparison is significant except for the N2 DBT-1(S78) - CB4856 comparisons (Tukey's HSD p-value < 1.43E-6).
DOI: https://doi.org/10.7554/eLife.40260.049

The following source data and figure supplements are available for figure 4:

**Source data 1.** Metabolite measurements for the CB4856 and CB4856 allele replacement strains (Used for *Figure 4B* and *Figure 4—figure supplement 1–3*).
DOI: https://doi.org/10.7554/eLife.40260.062

**Source data 2.** Processed metabolite measurements for the CB4856 and CB4856 allele replacement strains (Used for *Figure 4B* and *Figure 4—figure supplement 1–3*).
DOI: https://doi.org/10.7554/eLife.40260.063

**Source data 3.** Metabolite measurements for the N2 and N2 allele replacement strains (Used for *Figure 4B* and *Figure 4—figure supplement 1–3*).
DOI: https://doi.org/10.7554/eLife.40260.064

**Source data 4.** Processed phenotype data for the C15ISO rescue experiment (Used to generate *Figure 4C*).
DOI: https://doi.org/10.7554/eLife.40260.065

**Figure supplement 1.** Raw abundance of C17ISO for CB4856 and CB4856 allele replacement.
DOI: https://doi.org/10.7554/eLife.40260.050

**Figure supplement 2.** Straight-chain fatty acids are not affected by arsenic trioxide.
DOI: https://doi.org/10.7554/eLife.40260.051

**Figure supplement 2—source data 1.** C15ISO rescue trait loadings of principal components (PCs) for the PCs that explain up to 90% of the total variance in the trait data.
DOI: https://doi.org/10.7554/eLife.40260.052

**Figure supplement 3.** C15ISO and C17ISO to strait-chain ratios in control conditions.
DOI: https://doi.org/10.7554/eLife.40260.053

**Figure supplement 4.** Strains with the DBT-1(C78) allele produce more branched chain fatty acids in the L1 larval stage in control conditions.
DOI: https://doi.org/10.7554/eLife.40260.054

**Figure supplement 4—source data 1.** Metabolite measurements for N2, CB4856, and both allele-replacement strains at the L4 larval stage.
DOI: https://doi.org/10.7554/eLife.40260.055

**Figure supplement 5.** Young adult C15ISO and C17ISO to strait-chain ratios in control conditions.
DOI: https://doi.org/10.7554/eLife.40260.056

**Figure supplement 5—source data 1.** Trait correlations for the for the C15ISO rescue experiment (Used to generate *Figure 4—figure supplement 5*).
DOI: https://doi.org/10.7554/eLife.40260.057

**Figure supplement 6.** Complete results from C15iso rescue experiment. Tukey box plots for the PC1 trait after 0.64 μM C15ISO, arsenic trioxide, or arsenic trioxide and 0.64 μM C15ISO exposure are shown (N2, orange; CB4856, blue; allele replacement strains, gray).
DOI: https://doi.org/10.7554/eLife.40260.058

**Figure supplement 7.** BIOSORT-quantified traits for C15ISO rescue experiment Tukey box plots for the mean normalized optical density.
DOI: https://doi.org/10.7554/eLife.40260.059

**Figure supplement 8.** Trait correlations and principal component loadings of C15ISO rescue experiment.
DOI: https://doi.org/10.7554/eLife.40260.060

**Figure supplement 9.** Brood size and animal length are correlated with the first principal component for the C15ISO rescue experiment.
DOI: https://doi.org/10.7554/eLife.40260.061

together, these results suggest that the DBT-1(C78) allele produces more branched chain fatty acids than the DBT-1(S78) allele, but this effect was dependent on the developmental stage of the animals.

To test the hypothesis that differential arsenic-induced depletion of branched-chain fatty acids in strains with the DBT-1(C78S) causes physiological differences in growth, we tested if mmBCFA supplementation could rescue the effects of arsenic trioxide-induced toxicity. We exposed the parental and the DBT-1 allele-replacement strains to media containing arsenic trioxide alone, C15ISO alone, or a combination of arsenic trioxide and C15ISO. In agreement with previous experiments, the PC1 trait was more strongly correlated with the animal length, optical density, and fluorescence traits than the brood size trait (*Figure 4—figure supplement 6–7*; *Figure 4—source data 4*, *Figure 4—figure supplement 5—source data 1* and *Figure 4—figure supplement 2—source data 1*).

C15ISO supplementation of the arsenic growth media caused a 53.5% rescue of the allele-specific effect in the N2 genetic background (*Figure 4—figure supplement 6*). Similarly, when arsenic-exposed CB4856 DBT-1(C78) animals were supplemented with C15ISO, the allele-specific PC1 phenotypic difference was reduced by 25.6% when compared to the difference between the CB4856 DBT-1(C78) and CB4856 DBT-1(S78) strains in arsenic trioxide alone (*Figure 4—figure supplement 6C*). By contrast, CB4856 DBT-1(S78) and N2 DBT-1(S78) phenotypes were unaffected by C15ISO supplementation in arsenic trioxide media. We observed similar trends for the animal length, optical density, and fluorescence traits that we used as inputs for PCA but not for brood size (*Figure 4—figure supplement 9C*). Collectively, these data support the hypothesis that the cysteine/serine variant in DBT-1 contributes to arsenic sensitivity in *C. elegans* by reducing ISO fatty acid biosynthesis, and the DBT-1(C78) variant can be partially rescued by supplementation with mmBCFAs.

## Arsenic exposure increases mmBCFA production and favors a cysteine allele in human DBT1

To test whether our results from *C. elegans* translate to human variation in arsenic sensitivity, we tested the role of DBT1 variation on arsenic trioxide responses and mmBCFA biosynthesis in human cells. The human DBT1 enzyme contains a serine at position 112 that corresponds to the C78 residue in *C. elegans* DBT-1 (*Figure 4—figure supplement 6*). Using CRISPR-Cas9, we edited batch cultures of 293 T cells to generate a subset of cells with DBT1(S112C). These cells were exposed to arsenic trioxide or control conditions, and we monitored changes in the fraction of cells carrying the DBT1 (C112) allele. We found that arsenic exposure caused a 4% increase in the fraction of cells that contained the DBT1(C112) allele (*Figure 5B*, Fisher's exact test, p-value<1.9E-16; *Figure 5—source datas 1–2*). Though the human DBT1 does not vary within the human population at S112, two residues in close spatial proximity to S112 do vary among individuals in the population (R113C and W84C) (*Figure 5A*) (*Forbes et al., 2008*). To test the effects of these residues on arsenic trioxide sensitivity, we performed the same editing and arsenic selection procedure described above. Over the course of the selection experiment, cells with DBT1(W84C) and DBT1(R113C) increased by 2% and 1%, respectively (*Figure 5B*). Therefore, it appears that all three missense variants caused a slight increase in fitness in batch-edited human cell cultures exposed to arsenic – the opposite result we found in *C. elegans*. To determine if branched-chain fatty acid production was altered by arsenic exposure, as we found in *C. elegans*, we measured mmBCFA production in unedited 293 T cells in arsenic and mock-treated cultures. We found that overall fatty acid production was markedly reduced in arsenic-treated cultures. In contrast to our observations in *C. elegans*, straight-chain fatty acids were more drastically reduced than ISO fatty acids (*Figure 5—source data 3*), suggesting pleiotropic effects and a general perturbation of fatty acid metabolism.

## Discussion

In this study, we characterized the effects of *C. elegans* natural genetic variation on physiological responses to the pervasive environmental toxin arsenic trioxide. Though the effects of this toxin have been extensively studied in a variety of systems (*Ratnaike, 2003*; *Mandal and Suzuki, 2002*; *Bergquist et al., 2009*; *Paul et al., 2014*; *Shen et al., 2013*), recent evidence from human population studies have revealed local adaptations within region-specific subpopulations (*Schlebusch et al., 2015*; *Fujihara et al., 2009*; *Gomez-Rubio et al., 2010*; *Li et al., 2017*). Our investigation into the natural variation in *C. elegans* responses to arsenic trioxide led to the discovery of a novel mechanism by which this compound could elicit toxicity. We show that arsenic trioxide differentially inhibits two natural alleles of the E2 domain of the BCKDH complex, which is encoded by the *dbt-1* gene. Specifically, strains with the DBT-1(C78) allele are more sensitive to arsenic trioxide than strains carrying the DBT-1(S78). Furthermore, we show that the increased sensitivity of the DBT-1(C78) allele is largely explained by differences in the production of mmBCFAs (*Figure 4B–C*), which are critical molecules for developmental progression beyond the first larval stage. Arsenic is thought to inhibit the activity of both the pyruvate dehydrogenase (PDH) and the α-ketoglutarate (KGDH) dehydrogenase complexes through interactions with the reduced form of lipoate (*Bergquist et al., 2009*), which is the cofactor for the E2 domain of these complexes. Like the PDH and KGDH complexes, the E2 domain of BCKDH complex requires the cofactor lipoate to perform its enzymatic function (*Pettit et al., 1978*; *Heffelfinger et al., 1983*; *Yeaman, 1989*). The inhibition

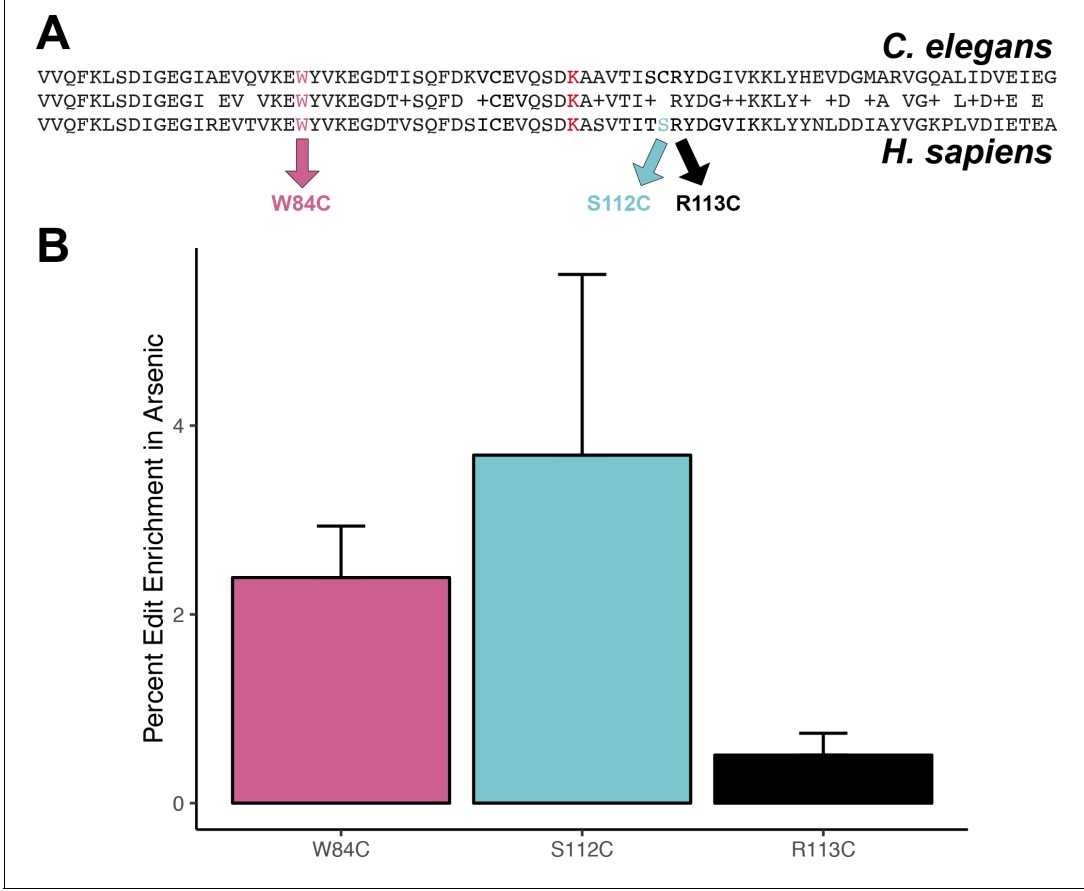

**Figure 5.** Protective effect of cysteine residues in human DBT1. (**A**) Alignment of *C. elegans* DBT-1 and *H. sapiens* DBT1. The residues tested for an arsenic-specific effect are indicated with arrows - W84C (pink), S112C (blue), and R113C (black). The lysine that is post-translationally modified with a lipoid acid is highlighted in red. (**B**) The percent increase of edited human cells that contain the W84C, S112C, or R113C amino acid change in DBT1 in the presence 5 µM arsenic trioxide relative to control conditions are shown. The number of reads in 5 µM arsenic trioxide for all replicates are significantly different from control conditions (Fisher's exact test, p-value<0.011).

DOI: https://doi.org/10.7554/eLife.40260.066

The following source data and figure supplements are available for figure 5:

**Source data 1.** Human cell line read data for CRISPR replacement experiment in 293 T cells.

DOI: https://doi.org/10.7554/eLife.40260.069

**Source data 2.** Results from Fisher's exact test of human cell line read data for CRISPR replacement experiment in 293 T cells.

DOI: https://doi.org/10.7554/eLife.40260.070

**Source data 3.** Metabolite measurements from human cell line experiments.

DOI: https://doi.org/10.7554/eLife.40260.071

**Figure supplement 1.** Three-dimensional homology model of *C. elegans* DBT-1 A three-dimensional homology model of *C. elegans* DBT-1 (black) aligned to human pyruvate dehydrogenase lipoyl domain (PDB:1Y8N) is shown.

DOI: https://doi.org/10.7554/eLife.40260.067

**Figure supplement 1—source data 1.** Tajima's D of GWA mapping confidence interval.

DOI: https://doi.org/10.7554/eLife.40260.068

of DBT-1 by arsenic trioxide could involve three-point coordination of arsenic by the C78 thiol and the reduced thiol groups of the nearby lipoate. However, based on the crystal structure (PDB:1Y8N), the atomic distance between the C78 thiol group and the thiol groups from the lipoylated lysine is ~32 Å, which might be too far for coordinating arsenic (*Figure 5—figure supplement 1*) (*Kato et al., 2005*). Alternatively, arsenic trioxide could inhibit DBT-1(C78) through coordination between the thiol groups of C78 and C65 (~8.5 Å) (*Figure 5—figure supplement 1*). Analogous thiol-dependent mechanisms have been proposed for the inhibition of other enzymes by arsenic

(*Shen et al., 2013*). Despite structural similarities and a shared cofactor, no evidence in the literature indicates that BCKDH is inhibited by arsenic trioxide, so these results demonstrate the first connection of arsenic toxicity to BCKDH E2 subunit inhibition.

Multiple sequence alignments show that cysteine residues C65 and C78 of *C. elegans* DBT-1 correspond to residues S112 and C99 of human DBT1 (*Figure 5A*). Although DBT1 does not vary at position 112 within the human population, two residues (R113C and W84C) in close spatial proximity do (*Forbes et al., 2008*). We hypothesized that cysteine variants in DBT1 would sensitize human cells to arsenic trioxide. However, we found that the cysteine variants (W84C, S112C, and R113C) proliferated slightly more rapidly than the parental cells in the presence of arsenic. Notably, a growing body of evidence suggests that certain cancer cells upregulate components involved in BCAA metabolism, and this upregulation promotes tumor growth (*Burrage et al., 2014*; *Tönjes et al., 2013*). Perhaps, the increased proliferation of human cell lines that contain the DBT1 C112 allele (*Figure 5*) is caused by increased activity of the BCKDH complex. It is worth noting that human cell lines grown in culture do not have the same strict requirements for mmBCFA, and the requirements for different fatty acids are variable among diverse immortalized cell lines (*Hughes-Fulford et al., 2001*; *Agostini et al., 2004*). Furthermore, in *C. elegans*, the developmental defects associated with *dbt-1* loss-of-function can be rescued by neuronal-specific expression of *dbt-1* (*Jia et al., 2016*), suggesting that the physiological requirements of mmBCFA in *C. elegans* depend on the coordination of multiple tissues that cannot be recapitulated with cell-culture experiments. These results further highlight the complexity of arsenic toxicity, as well as the physiological requirements for BCAA within and across species and could explain the discrepancy between the physiological effects we observed in *C. elegans* and human cell-line experiments. Given that arsenic trioxide has become the standard-of-care for treating APL (*Coombs et al., 2015*) and is gaining traction in treating other leukemias, it might be important to further explore the effects of arsenic on BCAA metabolism and cancer growth.

The C78 allele of DBT-1 is likely the derived allele in the *C. elegans* population because all other organisms have a serine at the corresponding position. The loss of the serine allele in the *C. elegans* population might have been caused by relaxed selection at this locus, although this hypothesis is difficult to test because of the effects of linked selection and decreased recombination in chromosome centers. It is hypothesized that the *C. elegans* species originated from the Pacific Rim and that the ancestral state more closely resembles the CB5846 strain than the N2 strain (*Andersen et al., 2012*; *Thompson et al., 2015*). The CB4856 strain was isolated from the Hawaiian island of O'ahu (*Hodgkin and Doniach, 1997*), where environments could have elevated levels of arsenic in the soil from volcanic activity, the farming of cane sugar, former construction material (canec) production facilities, or wood treatment plants (Hawaii.gov). It is possible that as the *C. elegans* species spread across the globe into areas with lower levels of arsenic in the soil and water, the selective pressure to maintain high arsenic tolerance was relaxed and the cysteine allele appeared. Alternatively, higher mmBCFA levels at the L1 larval stage in strains with the DBT-1(C78) allele (*Figure 4—figure supplements 3–4*) might cause faster development in certain conditions, although we did not observe allele-specific growth differences in laboratory conditions. Despite these clues suggesting selection in local environments, the genomic region surrounding the *dbt-1* locus does not show a signature of selection as measured by Tajima's D (*Tajima, 1989*) (*Figure 2—figure supplement 3*; *Figure 2—source data 3*), and the strains with the DBT-1 S78 allele show no signs of geographic clustering (*Figure 2—figure supplement 4*; *Figure 2—figure supplement 3—source data 1*). Nevertheless, our study suggests that *C. elegans* is a powerful model to investigate the molecular mechanisms for how populations respond to environmental toxins.

## Materials and methods

### Key resources table

| Reagent type (species) or resource | Designation | Source or reference | Identifiers | Additional information |
|---|---|---|---|---|

*Continued on next page*

*Continued*

| Reagent type (species) or resource | Designation | Source or reference | Identifiers | Additional information |
|---|---|---|---|---|
| Gene (*Caenorhabditis elegans*) | *dbt-1* | NA | Wormbase: WBGene00014054 | |
| Strain, strain background (*C. elegans*) | N2 DBT-1(S78) | This paper | Andersen_Lab:ECA581 | *dbt-1(ean15*[C78S]) |
| Strain, strain background (*C. elegans*) | CB4856 DBT-1(C78) | This paper | Andersen_Lab:ECA590 | *dbt-1(ean34*[S78C]) |
| Strain, strain background (*C. elegans*) | Left NIL; CB4856 > N2 (II:5.75–8.02 Mb) | This paper | Andersen_Lab:ECA414 | *eanIR188*[II:5.75–8.02 Mb, CB4856 > N2] |
| Strain, strain background (*C. elegans*) | Right NIL; CB4856 > N2 (II:7.83–9.66 Mb) | This paper | Andersen_Lab:ECA434 | *eanIR208*[II:7.83–9.66 Mb, CB4856 > N2] |
| Sequence-based reagent | NIL Fd primer | This paper | Andersen_Lab:oECA609 | tttcacacaaaccatgcgct |
| Sequence-based reagent | NIL Rv primer | This paper | Andersen_Lab:oECA610 | actcgtctgctgggtattct |
| Sequence-based reagent | NIL Fd primer | This paper | Andersen_Lab:oECA611 | tgtcttcgcacctttactcg |
| Sequence-based reagent | NIL Rv primer | This paper | Andersen_Lab:oECA612 | cattcaagtcagggcgtatcc |
| Sequence-based reagent | Genotype C78S Edit | This paper | Andersen_Lab:oECA1163 | GAAGGAATTGC CGAAGTTCAGGTTAAG |
| Sequence-based reagent | Genotype C78S Edit | This paper | Andersen_Lab:oECA1165 | CCGTCATCTCCAC AAAAAGCTTTATCTCTC |
| Sequence-based reagent | *dbt-1* gRNA | This paper | Andersen_Lab:crECA97 | CCATCTCCTGTAGATACGAC |
| Sequence-based reagent | N2 *dbt-1* repair oligo | This paper | Andersen_Lab:oECA1542 | CTTCCAGGTACGTGA AAGAAGGAGATACGATT TCGCAGTTCGATAAAGTCTGT GAAGTGCAAAGTGATAAAGCAGC AGTAACCATCTCCAGTAGATACGA CGGAATTGTCAAAAAATTG TAAGTTTCTTCCTAA |
| Sequence-based reagent | CB4856 *dbt-1* repair oligo | This paper | Andersen_Lab:oECA1543 | TTAGGAAGAAACTTACAATT TTTTGACAATTCCGTCGTA TCTACAGGAGATGGTTACTGCT GCTTTATCGCTTTGCACTTCACAG ACTTTATCGAAC TGCGAAATCGTA TCTCCTTCTTTCA CGTACCTGGAAG |
| Sequence-based reagent | *dpy-10* repair oligo | *Kim et al., 2014* | Andersen_Lab:crECA37 | CACTTGAACTTCAATACGGCAA GATGAGAATGACT GGAAACCGTACCGC ATGCGGTGCCTATGGTAGCGGA GCTTCACATGGCT TCAGACCAACAGCCTAT |
| Sequence-based reagent | *dpy-10* gRNA | *Kim et al., 2014* | Andersen_Lab:crECA36 | GCTACCATAGGCACCACGAG |
| Sequence-based reagent | Human gRNA S112C and R113C | This paper | Guide_1 used in RDA_74 | TCCATCATAACGACTAGTGA |
| Sequence-based reagent | S112C repair template | This paper | 1192 DBT1-repair-S112C | ATAGCATCTGTGAAGTTCA AAGTGATAAAGCTTCTGTTAC AATCACTTGTCGTTATGATGGA GTCATTAAAAAACTCTATT |

*Continued on next page*

*Continued*

| Reagent type (species) or resource | Designation | Source or reference | Identifiers | Additional information |
|---|---|---|---|---|
| Sequence-based reagent | R113C repair template | This paper | 1193 DBT1-repair-R113C | ATAGCATCTGTGAAGTT CAAAGTGATAAAGCTTCTG TTACAATCACTAGTTGTTAT GATGGAGTCATTAAAAAACTCTATT |
| Sequence-based reagent | Fwd PCR C | This paper | 1188 DBT1-PCR-C | Ttgtggaaaggacgaaacacc gAGAAGGAGATACAGTGTCTCAGT |
| Sequence-based reagent | Fwd PCR D | This paper | 1189 DBT1-PCR-D | Ttgtggaaaggacgaaa caccgTGTCTCAGTTTG ATAGCATCTGTG |
| Sequence-based reagent | Human gRNA W84C | This paper | Guide_2 used in RDA_75 | TCTTTTAGGTAT GTAAAAGA |
| Sequence-based reagent | W84C repair template | This paper | 1195 DBT1-repair-W84C-v2 | GACTGTTTCCATAAAA GTGTCTCATTTCTTTT TCTTTTAGTTATGTGAAG GAAGGAGATACAGTGTCTCAG TTTGATAGCAT |
| Sequence-based reagent | Fwd PCR A | This paper | 1186 DBT1-PCR-A | Ttgtggaaaggacgaaacac cgGCATGGCATTTACATC CTTAATATGAT |
| Sequence-based reagent | Fwd PCR B | This paper | 1187 DBT1-PCR-B | Ttgtggaaaggacgaaaca ccgCCTTAATATGATCTGT ACTTATGACTGTTT |
| Sequence-based reagent | Rev PCR 1 | This paper | 1190 DBT1-PCR-Rev1 | Tctactattctttcccctgca ctgtCTACTAATGGCTTCCCCACAT |
| Sequence-based reagent | Rev PCR 2 | This paper | 1191 DBT1-PCR-Rev2 | Tctactattctttcccctgcactgt CAATACCTTTTAAAGC TTCCGTTTCTAT |
| Transfected construct (*Homo Sapiens*) | S112C and R113C Cas9-sgRNA plasmid | This paper | p1054 | |
| Transfected construct (*Homo Sapiens*) | W84C Cas9-sgRNA plasmid | This paper | p1052 | |

## Strains

Animals were fed the bacterial strain OP50 and grown at 20℃ on modified nematode growth medium (NGM), containing 1% agar and 0.7% agarose to prevent burrowing of the wild isolates (*Boyd et al., 2012*). For each assay, strains were grown for five generations with no strain entering starvation or encountering dauer-inducing conditions (*Andersen et al., 2014*). Wild *C. elegans* isolates used for genome-wide association and recombinant inbred advanced intercross lines (RIAILs) used for linkage mapping have been described previously (*Cook et al., 2016*; *Cook et al., 2017*; *Andersen et al., 2015*). Strains constructed for this manuscript are listed above in the Key Resources Table.

## High-throughput fitness assay

We used the high-throughput fitness assay (HTA) described previously (*Andersen et al., 2015*). In short, strains are passaged for four generations before bleach-synchronization and aliquoted to 96-well microtiter plates at approximately one embryo per microliter in K medium (*Boyd et al., 2012*). The final concentration of NaCl in the K medium for the genome-wide association (GWA) and linkage mapping assays was 51 mM. For all subsequent experiments the final NaCl concentration was 10.2 mM. The following day, hatched and synchronized L1 animals were fed HB101 bacterial lysate (Pennsylvania State University Shared Fermentation Facility, State College, PA, [*García-González et al., 2017*]) at a final concentration of 5 mg/ml and grown for two days at 20℃. Next, three L4 larvae were sorted using a large-particle flow cytometer (COPAS BIOSORT, Union Biometrica, Holliston, MA) into microtiter plates that contain HB101 lysate at 10 mg/ml, K medium, 31.25

µM kanamycin, and either arsenic trioxide dissolved in 1% water or 1% water alone. The animals were then grown for four days at 20°C. For linkage mapping and GWA mapping experiments, we added polychromatic fluorescent beads (Polysciences, cat. #19507–5) to each well for five minutes. The populations were treated with sodium azide (50 mM) prior to being measured with the BIO-SORT. To reduce experimental costs, the polychromatic fluorescent beads were not added to follow-up experiments. For all experiments, we report the results for four independently quantified traits): the normalized brood size (norm.n), mean progeny length per well (mean.TOF), the mean optical density normalized by animal length per well (mean.norm.EXT), and the mean fluorescence normalized by animal length per well (mean.norm.yellow). All raw experimental data can be found on FigShare (https://doi.org/10.6084/m9.figshare.7458980.v2).

## Calculation of fitness traits for genetic mappings

Phenotype data generated using the BIOSORT were processed using the R package *easysorter*, which was specifically developed for processing this type of data (*Shimko and Andersen, 2014*). Briefly, the function *read_data*, reads in raw phenotype data and runs a support vector machine to identify and eliminate bubbles. Next, the *remove_contamination* function eliminates any wells that were identified as contaminated prior to scoring population parameters. This analysis results in processed BIOSORT data where each observation is for a given strain corresponds to the measurements for an individual animal. However, the phenotypes we used for mapping and follow-up experiments are summarized statistics of populations of animals in each well of a 96-well plate. The *sumplate* function was used to generate summary statistics of the measured parameters for each animal in each well. These summary statistics include the 10th, 25th, 50th, 75th, and 90th quantiles for time of flight (TOF), animal extinction (EXT), and three fluorescence channels (Green, Yellow, and Red), which correspond to animal length, optical density, and ability to pump fluorescent beads, respectively. Measured brood sizes (n) are normalized by the number of animals that were originally sorted into each well (norm.n). For mapping experiments, a single well replicate for each strain is summarized using the *sumplate* function. For follow-up experiments, multiple replicates for each strain indicated by a unique plate, well, and column were used. After summary statistics for each well are calculated, we accounted for differences between assays using the *regress(assay = TRUE)* function in the *easysorter* package. Outliers in the GWA and linkage mapping experiments were identified and eliminated using the *bamf_prune* function in *easysorter*. For follow-up experiments that contained multiple replicates for each strain, we eliminated strain replicates that were more than two standard deviations from the strain mean for each condition tested. Finally, arsenic-specific effects were calculated using the *regress(assay = FALSE)* function from *easysorter*, which accounts for strain-specific differences in growth parameters present in control conditions.

## Principal component analysis of processed BIOSORT measured traits

The COPAS BIOSORT measures individual animal length (TOF), optical density (EXT), fluorescence (green, yellow, and red). We use these data to calculate the total number of animals in a well and then normalize by the number of animals initially sorted into the well (brood size). All these measurements were then summarized using the *easysorter* package to generate various summary statistics of each measured parameter, including five distribution quantiles and measures of dispersion (*Andersen et al., 2015*). We used four independently quantified traits as inputs to principal component analysis (PCA): the normalized brood size (norm.n), mean progeny length per well (mean.TOF), the mean optical density normalized by animal length per well (mean.norm.EXT), and the mean fluorescence normalized by animal length per well (mean.norm.yellow). Although we only used fluorescent beads in the GWA and linkage mapping experiments, we found that fluorescence-based traits exhibited an arsenic-specific effect that correlated with strain sickness. Prior to principal component analysis (PCA), HTA phenotypes were scaled to have a mean of zero and a standard deviation of one using the *scale* function in R. PCA was performed using the *prcomp* function in R (*R Development Core Team, 2017*). Eigenvectors were subsequently extracted from the object returned by the *prcomp* function.

## Arsenic trioxide dose-response assays

All dose-response experiments were performed on four genetically diverged strains (N2, CB4856, DL238, and JU775) in technical quadruplicates prior to performing GWA and linkage mapping experiments (*Figure 1—source data 1*). Animals were assayed using the HTA, and phenotypic analyses were performed as described above. The arsenic trioxide concentration for GWA and linkage mapping experiments was chosen based on an observable effect for animal length and brood size phenotypes in the presence of arsenic.

## Heritability calculations

For dose-response experiments, broad-sense heritability ($H^2$) estimates were calculated using the *lmer* function in the lme4 package with the following linear mixed model (phenotype ~1 + (1|strain)) (*Bates et al., 2014*). $H^2$ was then calculated as the fraction of the total variance that can be explained by the random component (strain) of the mixed model. Prior to estimating $H^2$, we removed outlier replicates that we defined as replicates with values greater than two standard deviations away from the mean phenotype. Outliers were defined on a per-trait and per-strain basis. Heritability estimates for dose response experiments are in *Figure 1—figure supplement 3—source data 1*.

Heritability estimates for the linkage mapping experiment were calculated using two approaches. In both approaches, we used the previously described RIAIL genotype matrix to compute relatedness matrices (*Andersen et al., 2012*). In the first approach, a variance component model using the R package regress was used to estimate the fraction of phenotypic variation explained by additive and epistatic genetic factors, $H^2$, or just additive genetic factors, $h^2$ (*Bloom et al., 2015*; *David Clifford, 2006*), using the formula (y ~ 1, ~ZA+ZAA), where y is a vector of RIAIL phenotypes, ZA is the additive relatedness matrix, and ZAA is the pairwise-interaction relatedness matrix. The additive relatedness matrix was calculated as the correlation of marker genotypes between each pair of strains. In addition, a two-component variance model was calculated with both an additive and pairwise-interaction effect. The pairwise-interaction relatedness matrix was calculated as the Hadamard product of the additive relatedness matrix.

The second approach utilized a linear mixed model and the realized additive and epistatic relatedness matrices (*Endelman, 2011*; *Covarrubias-Pazaran, 2016*; *Su et al., 2012*; *Endelman and Jannink, 2012*). We used the *mmer* function in the sommer package with the formula (y ~ A + E) to estimate variance components, where y is a vector of RIAIL phenotypes, A is the realized additive relatedness matrix, and E is the epistatic relatedness matrix. This same approach was used to estimate heritability for the GWA mapping phenotype data, with the only difference being that we used the wild isolate genotype matrix described below. Heritability estimates for RIAIL and wild isolate data are in *Figure 1—figure supplement 6—source data 1* and *Figure 2—figure supplement 5—source data 1*, respectively.

## Effect size calculations for dose response assay

We first fit a linear model with the formula (phenotype ~strain) for all measured and principal component traits for each concentration of arsenic trioxide using the *lm* R function. Next, we extracted effect sizes using the *anova_stats* function from the sjstats R package (*Lüdecke, 2018*). Effect sizes for dose responses are in *Figure 1—figure supplement 3—source data 2*.

## Linkage mapping

A total of 262 RIAILs were phenotyped in the HTA described previously for control and arsenic trioxide conditions (*Andersen et al., 2015*; *Zdraljevic et al., 2017*). The phenotype and genotype data were entered into R and scaled to have a mean of zero and a variance of one for linkage analysis (*Figure 1—source data 2*). Quantitative trait loci (QTL) were detected by calculating logarithm of odds (LOD) scores for each marker and each trait as $-n(ln(1 - r^2)/2ln(10))$, where $r$ is the Pearson correlation coefficient between RIAIL genotypes at the marker and phenotype trait values (*Bloom et al., 2013*). The maximum LOD score for each chromosome for each trait was retained from three iterations of linkage mappings (*Figure 1—source data 3*). We randomly permuted the phenotype values of each RIAIL while maintaining correlation structure among phenotypes 1000 times to estimate the significance threshold empirically. The significance threshold was set using a

genome-wide error rate of 5%. Confidence intervals were defined as the regions contained within a 1.5 LOD drop from the maximum LOD score (*Broman et al., 2003*).

## Near-isogenic line (NIL) generation

NILs were generated by crossing N2xCB4856 RIAILs to each parental genotype. For each NIL, eight crosses were performed followed by six generations of selfing to homozygose the genome. Reagents used to generate NILs are detailed in the Key Resources Table. The NILs responses to 1000 μM arsenic trioxide were quantified using the HTA described above (*Figure 1—source data 4*). NIL whole-genome sequencing and analysis was performed as described previously (*Brady et al., 2018*) (*Figure 1—source data 5*).

## Genome-wide association mapping

Genome-wide association (GWA) mapping was performed using phenotype data from 86 *C. elegans* isotypes (*Figure 2—source data 2*). Genotype data were acquired from the latest VCF release (Release 20180527) from CeNDR that was imputed as described previously (*Cook et al., 2017*). We used BCFtools (*Li, 2011*) to filter variants that had any missing genotype calls and variants that were below 5% minor allele frequency. We used PLINK v1.9 (*Purcell et al., 2007*; *Chang et al., 2015*) to LD-prune the genotypes at a threshold of $r^2$ <--indep-pairwise 50 10 0.8.0.8, using *–indep-pairwise 50 10 0.8.* This resulting genotype set consisted of 59,241 markers that were used to generate the realized additive kinship matrix using the *A.mat* function in the *rrBLUP* R package (*Endelman, 2011*) (*Figure 2—source data 3*). These markers were also used for genome-wide mapping. However, because these markers still have substantial LD within this genotype set, we performed eigen decomposition of the correlation matrix of the genotype matrix using *eigs_sym* function in Rspectra package (*Qiu, 2018*). The correlation matrix was generated using the *cor* function in the correlateR R package (*Bilgrau, 2018*). We set any eigenvalue greater than one from this analysis to one and summed all of the resulting eigenvalues (*Li and Ji, 2005*). This number was 500.761, which corresponds to the number of independent tests within the genotype matrix. We used the *GWAS* function in the rrBLUP package to perform genome-wide mapping with the following command: *rrBLUP::GWAS* (*pheno = PC1, geno = Pruned_Markers, K = KINSHIP, min.MAF = 0.05, n.core = 1, P3D = FALSE, plot = FALSE*). To perform fine-mapping, we defined confidence intervals from the genome-wide mapping as +/- 100 SNVs from the rightmost and leftmost markers above the Bonferroni significance threshold. We then generated a QTL region of interest genotype matrix that was filtered as described above, with the one exception that we did not perform LD pruning. We used PLINK v1.9 to extract the LD between the markers used for fine mapping and the peak QTL marker identified from the genome-wide scan. We used the same command as above to perform fine mapping, but with the reduced variant set. The workflow for performing GWA mapping can be found on https://github.com/AndersenLab/cegwas2-nf (copy archived at https://github.com/elifesciences-publications/cegwas2-nf; *Zdraljevic et al., 2019*). All trait mapping results can be found on FigShare (https://doi.org/10.6084/m9.figshare.7828706.v1).

## Generation of *dbt-1* allele replacement strains

Allele replacement strains were generated using CRISPR-Cas9-mediated genome editing, using the co-CRISPR approach (*Kim et al., 2014*) with Cas9 ribonucleoprotein delivery (*Paix et al., 2015*). Alt-R crRNA and tracrRNA were purchased from IDT (Skokie, IL). tracrRNA (IDT, 1072532) was injected at a concentration of 13.6 μM. The *dpy-10* and the *dbt-1* crRNAs were injected at 4 μM and 9.6 μM, respectively. The *dpy-10* and the *dbt-1* single-stranded oligodeoxynucleotides (ssODN) repair templates were injected at 1.34 μM and 4 μM, respectively. Cas9 protein (IDT, 1074182) was injected at 23 μM. To generate injection mixes, the tracrRNA and crRNAs were incubated at 95°C for 5 min and 10°C for 10 min. Next, Cas9 protein was added and incubated for 5 min at room temperature. Finally, repair templates and nuclease-free water were added to the mixtures and loaded into pulled injection needles (1B100F-4, World Precision Instruments, Sarasota, FL). Individual injected $P_0$ animals were transferred to new 6 cm NGM plates approximately 18 hr after injections. Individual $F_1$ rollers were then transferred to new 6 cm plates to generate self-progeny. The region surrounding the desired S78C (or C78S) edit was then amplified from $F_1$ rollers using primers oECA1163 and oECA1165. The PCR products were digested using the *Sfc*I restriction enzyme

(R0561S, New England Biolabs, Ipswich, MA). Differential band patterns signified successfully edited strains because the N2 S78C, which is encoded by the CAG codon, creates an additional *Sfc*I cut site. Non-Dpy, non-Rol progeny from homozygous edited $F_1$ animals were propagated. If no homozygous edits were obtained, heterozygous $F_1$ progeny were propagated and screened for the presence of the homozygous edits. $F_1$ and $F_2$ progeny were then Sanger sequenced to verify the presence of the proper edited sequence. The phenotypes of allele replacement strains in control and arsenic trioxide conditions were measured using the HTA described above (*Figure 1—source data 4*). PCA phenotypes for allele-replacement strains were generated the same way as described above for GWA mapping traits and are located in *Figure 1—source data 4*.

## Rescue with 13-methyltetradecanoic acid

Strains were grown as described for a standard HTA experiment. In addition to adding arsenic tri-oixde to experimental wells, we also added a range of C15iso (13-methyltetradecanoic acid, Matreya Catalog # 1605) concentrations to assay rescue of arsenic effects (*Figure 4—source data 4*).

## Growth conditions for metabolite profiling

For L1 larval stage assays, chunks (~1 cm) were taken from starved plates and placed on multiple fresh 10 cm plates. Prior to starvation, animals were washed off of the plates using M9, and embryos were prepared by bleach synchronization. Approximately 40,000 embryos were resuspended in 25 ml of K medium and allowed to hatch overnight at 20°C. L1 larvae were fed 15 mg/ml of HB101 lysate the following morning and allowed to grow at 20°C for 72 hr. We harvested 100,000 embryos from gravid adults by bleaching. These embryos were hatched overnight in 50 ml of K medium in a 125 ml flask. The following day, we added arsenic trioxide to a final concentration of 100 µM and incubated the cultures for 24 hr. After 24 hr, we added HB101 bacterial lysate (2 mg/ml) to each culture. Finally, we transferred the cultures to 50 ml conical tubes, centrifuged the cultures at 3000 RPM for 3 min to separate the pellet and supernatant. The supernatant and pellets from the cultures were frozen at −80°C and prepared for analysis. For young adult stage assays, 45,000 animals per culture were prepared as described above but in S medium, at a density of three animals per microliter, and fed HB101 lysate (5 mg/mL). These cultures were shaken at 200 RPM, 20°C in 50 mL Erlenmeyer flasks for 62 hr. For harvesting, we settled 15 mL of cultures for 15 min at room temperature and then pipetted the top 12 mL of solution off of the culture. The remaining 3 mL of animal pellet was washed with 10 mL of M9, centrifuged at 1000 g for one minute, and then the supernatant removed. This wash was repeated once more with M9 and again with water. The final nematode pellet was snap frozen in liquid nitrogen.

## Nematode metabolite extractions

Pellets were lyophilized 18–24 hr using a VirTis BenchTop 4K Freeze Dryer until a chalky consistency was achieved. Dried pellets were transferred to 1.5 mL microfuge tubes and dry pellet weight recorded. Pellets were disrupted in a Spex 1600 MiniG tissue grinder after the addition of three stainless steel grinding balls to each sample. Microfuge tubes were placed in a Cryoblock (Model 1660) cooled in liquid nitrogen, and samples were disrupted at 1100 RPM for two cycles of 30 s. Each sample was individually dragged across a microfuge tube rack eight times, inverted, and flicked five times to prevent clumping. This process was repeated two additional rounds for a total of six disruptions. Pellets were transferred to 4 mL glass vials in 3 mL 100% ethanol. Samples were sonicated for 20 min (on/off pulse cycles of two seconds at power 90 A) using a Qsonica Ultrasonic Processor (Model Q700) with a water bath cup horn adaptor (Model 431C2). Following sonication, glass vials were centrifuged at 2750 RCF for five minutes in an Eppendorf 5702 Centrifuge using rotor F-35-30-17. The resulting supernatant was transferred to a clean 4 mL glass vial and concentrated to dryness in an SC250EXP Speedvac Concentrator coupled to an RVT5105 Refrigerated Vapor Trap (Thermo Scientific). The resulting powder was suspended in 100% ethanol according to its original dry pellet weight: 0.01 mL 100% ethanol per mg of material. The suspension was sonicated for 10 min (pulse cycles of 2 s on and 3 s off at power 90 A) followed by centrifugation at 20,817 RCF in a refrigerated Eppendorf centrifuge 5417R at 4°C. The resulting supernatant was transferred to an HPLC vial containing a Phenomenex insert (cat #AR0-4521-12) and centrifuged at 2750 RCF for five

minutes in an Eppendorf 5702 centrifuge. The resulting supernatant was transferred to a clean HPLC vial insert and stored at −20°C or analyzed immediately.

## Mass spectrometric analysis

Reversed-phase chromatography was performed using a Dionex Ultimate 3000 Series LC system (HPG-3400 RS High Pressure pump, TCC-3000RS column compartment, WPS-3000TRS autosampler, DAD-3000 Diode Array Detector) controlled by Chromeleon Software (ThermoFisher Scientific) and coupled to an Orbitrap Q-Exactive mass spectrometer controlled by Xcalibur software (Thermo-Fisher Scientific). Metabolites were separated on a Kinetex EVO C18 column, 150 mm x 2.1 mm, particle size 1.7 μm, maintained at 40°C with a flow rate of 0.5 mL/min. Solvent A: 0.1% ammonium acetate in water; solvent B: acetonitrile (ACN). A/B gradient started at 5% B for 30 s, followed by a linear gradient to 95% B over 13.5 min, then a linear gradient to 100% B over 3 min. 100% B was maintained for 1 min. Column was washed after each run with 5:1 isopropanol:ACN, flow rate of 0.12 mL/min for 5 min, followed by 100% ACN for 2.9 min, a linear gradient to 95:5 water:ACN over 0.1 min, and then 95:5 water:ACN for 2 min with a flow rate of 0.5 mL/min. A heated electrospray ionization source (HESI-II) was used for the ionization with the following mass spectrometer parameters: spray voltage: 3 kV; capillary temperature: 320°C; probe heater temperature: 300°C; sheath gas: 70 AU; auxiliary gas flow: 2 AU; resolution: 240,000 FWHM at m/z 200; AGC target: 5e6; maximum injection time: 300 ms. Each sample was analyzed in negative and positive modes with m/z range 200–800. Fatty acids and most ascarosides were detected as [M-H]⁻ ions in negative ionization mode. Peaks of known abundant ascarosides and fatty acids were used to monitor mass accuracy, chromatographic peak shape, and instrument sensitivity for each sample. Processed metabolite measures can be found in *Figure 4—source datas 1–3* and *Figure 4—figure supplement 4—source datas 1* (*Artyukhin et al., 2018*).

## Statistical analyses

All p-values testing the differences of strain phenotypes in the NIL, allele-replacement, and C15ISO experiments were performed in R using the *TukeyHSD* function with an ANOVA model with the formula (*phenotype ~strain*). p-Values of individual pairwise strain comparisons are reported in each figure legend.

## CRISPR-Cas9 gene editing in human cells

The 293 T cells were sourced from CCLE. Identity authenticated by SNP profiling. Cells were regularly tested for Mycoplasma (~bimonthly). Gene-editing experiments were performed in a single parallel culture experiment using human 293 T cells (ATCC) grown in DMEM with 10% FBS. On day zero, 300,000 cells were seeded per well in a six-well plate format. The following day, two master mixes were prepared: a) LT-1 transfection reagent (Mirus) was diluted 1:10 in Opti-MEM and incubated for 5 min; b) a DNA mix of 500 ng Cas9-sgRNA plasmid (*Supplementary file 1–2*) with 250 pmol repair template oligonucleotide was diluted in Opti-MEM in a final volume of 100 μL. 250 μL of the lipid mix was added to each of the DNA mixes and incubated at room temperature for 25 min. Following incubation, the full 350 μL volume of DNA and lipid mix was added dropwise to the cells. These six-well plates were then centrifuged at 1000 x g for 30 min. After 6 hr, the media on the cells was replaced. For the next 6 days, cells were expanded and passaged as needed. On day 7, one million cells were taken from each set of edited and unedited cells and placed into separate T75s with either media-only or 5 μM arsenic-containing media. Days 7 to 14, arsenic and media-only conditions were maintained at healthy cell densities. Days 14 to 18, arsenic exposed cell populations were maintained off arsenic to allow the populations to recover prior to sequencing. Media-only conditions were maintained in parallel. On day 18, all arsenic and media-only conditions were pelleted for genomic DNA extraction.

## Analysis of CRISPR-Cas9 editing in human cells

Genomic DNA was extracted from cell pellets using the QIAGEN (QIAGEN, Hilden, Germany) Midi or Mini Kits based on the size of the cell pellet (51183, 51104) according to the manufacturer's recommendations. DBT1 loci were first amplified with 17 cycles of PCR using a touchdown protocol and the NEBnext 2x master mix (New England Biolabs M0541). The resulting product served as

input to a second PCR, using primers that appended a sample-specific barcode and the necessary adaptors for Illumina sequencing. The resulting DNA was pooled, purified with SPRI beads (A63880, Beckman Coulter, Brea, CA), and sequenced on an Illumina MiSeq with a 300-nucleotide single-end read with an eight nucleotide index read. For each sample, the number of reads exactly matching the wild-type and edited DBT1 sequence were determined (*Figure 5—source data 1*).

## Preparing human cells for mass spectroscopy

Mass spectroscopy experiments used human 293 T cells (ATCC) grown in DMEM with 10% FBS. On day zero, 150,000 cells were seeded into 15 cm tissue cultures dishes with 15 mL of either 2.5 μM arsenic or no arsenic media. Each condition had five replicates. On day 3, the no arsenic cells were approaching confluence and required passaging. Arsenic conditions were at ~30% confluence and received a media change. On day seven, both conditions were near confluence, media was removed, and plates were rinsed with ice cold PBS, remaining liquid removed. Plates were frozen at −80°C before processing for mass spectrometric analysis. Cells were scraped off the plates with PBS and pelleted in microfuge tubes. Cell pellets were lyophilized 18–24 hr using a VirTis BenchTop 4K Freeze Dryer and extracted in 100% ethanol using the same sonication program as described for nematode extraction. Following sonication, samples were centrifuged at 20,817 RCF in a refrigerated Eppendorf centrifuge 5417R at 4°C. Clarified supernatant was aliquoted to a new tube and concentrated to dryness in an SC250EXP Speedvac Concentrator coupled to an RVT5105 Refrigerated Vapor Trap (Thermo Scientific). The resulting material was suspended in. 1 mL 100% ethanol and analyzed by LC-MS as described. Metabolite measurements can be found in *Figure 5—source data 3*.

## Tajima's D calculation

We used the VCF corresponding to CeNDR release 20160408 (https://elegansvariation.org/data/release/20160408) to calculate Tajima's D. Tajima's D was calculated using the *tajimas_d* function in the *cegwas* package using default parameters (window size = 500 SNVs, sliding window distance = 50 SNVs, outgroup = N2) (*Figure 2—source data 3*). Isolation locations of strains can be found in *Figure 2—figure supplement 3—source data 1*.

# Acknowledgements

The authors thank Samuel Rosenberg for assistance on early mappings of drug sensitivities, Mudra Hegde of the Broad Institute for assistance with sequence analysis, and members of the Andersen laboratory for critical reading of this manuscript.

# Additional information

### Funding

| Funder | Grant reference number | Author |
|---|---|---|
| National Institute of General Medical Sciences | T32GM008061 | Stefan Zdraljevic |
| National Institute of Diabetes and Digestive and Kidney Diseases | DK115690 | Frank C Schroeder<br>Erik C Andersen |
| American Cancer Society | 127313-RSG-15-135-01-DD | Frank C Schroeder |
| National Institute of Environmental Health Sciences | ES029930 | Erik C Andersen |
| National Institute of General Medical Sciences | GM088290 | Erik C Andersen |
| Next Generation Fund | | Erik C Andersen |
| National Institute of Environmental Health Sciences | ES029930 | Erik C Andersen |

| Sherman-Fairchild Cancer Innovation Award | Erik C Andersen |

The funders had no role in study design, data collection and interpretation, or the decision to submit the work for publication.

## Author contributions

Stefan Zdraljevic, Conceptualization, Data curation, Formal analysis, Funding acquisition, Validation, Investigation, Visualization, Methodology, Writing—original draft, Project administration, Writing—review and editing; Bennett William Fox, Data curation, Formal analysis, Investigation, Methodology, Writing—review and editing; Christine Strand, Data curation, Formal analysis, Validation, Investigation, Writing—review and editing; Oishika Panda, Data curation, Formal analysis, Investigation; Francisco J Tenjo, Investigation; Shannon C Brady, Resources, Writing—review and editing; Tim A Crombie, Investigation, Writing—review and editing; John G Doench, Resources, Supervision, Methodology, Project administration, Writing—review and editing; Frank C Schroeder, Supervision, Funding acquisition, Methodology, Project administration, Writing—review and editing; Erik C Andersen, Conceptualization, Resources, Supervision, Funding acquisition, Methodology, Project administration, Writing—review and editing

## Author ORCIDs

Stefan Zdraljevic ⓘ http://orcid.org/0000-0003-2883-4616
John G Doench ⓘ http://orcid.org/0000-0002-3707-9889
Erik C Andersen ⓘ http://orcid.org/0000-0003-0229-9651

## Decision letter and Author response

Decision letter https://doi.org/10.7554/eLife.40260.080
Author response https://doi.org/10.7554/eLife.40260.081

# Additional files

## Supplementary files

• Supplementary file 1. Plasmid used for editing human cells with the S112C and R113C edits.
DOI: https://doi.org/10.7554/eLife.40260.072

• Supplementary file 2. Plasmid used for editing human cells with the W84C edit.
DOI: https://doi.org/10.7554/eLife.40260.073

• Transparent reporting form
DOI: https://doi.org/10.7554/eLife.40260.074

## Data availability

All data generated or analyzed during this study are included in the manuscript and supporting files.

The following previously published dataset was used:

| Author(s) | Year | Dataset title | Dataset URL | Database and Identifier |
|-----------|------|---------------|-------------|-------------------------|
| Cook DE, Andersen EC | 2016 | CeNDR wild isolate WGS data | https://www.ncbi.nlm.nih.gov/sra/?term=PRJNA318647 | NCBI Sequence Read Archive, PRJNA318647 |

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
