## [Decision Letter]

[**Editorial note:** This article has been through an editorial process in which the authors decide how to respond to the issues raised during peer review. The Reviewing Editor's assessment is that all the issues have been addressed.]

Thank you for submitting your article "Natural variation in arsenic toxicity is explained by differences in branched chain amino acid catabolism" for consideration by *eLife*. Your article has been reviewed by two peer reviewers, and the evaluation has been overseen by a Reviewing Editor and Michael Marletta as the Senior Editor. The following individual involved in review of your submission has agreed to reveal her identity: Meng C Wang (Reviewer #2).

The Reviewing Editor has highlighted the concerns that require revision and/or responses, and we have included the separate reviews below for your consideration. If you have any questions, please do not hesitate to contact us

Summary:

As you will see from the reviews, both reviewers appreciated the breadth of the work and the potential overall interest. The major strengths of the work are:

a) The identification of a new gene (DBT-1) in regulating arsenic toxicity in both *C. elegans* and mammalian cells.

b) The comprehensive experimental approach starting from analyses of natural variation in *C. elegans* strains to identification of a genetic contributor to this variation, followed by biochemical characterization of the encoded enzyme.

However, both reviewers also raised a number of issues that are summarized in brief below.

Major concerns:

The major concern is that it is unclear from the presented data to what extent variation in dbt-1 contributes to the observed genetic variation in arsenic toxicity. The methods and analyses (particularly for the GWAS) are either not presented or not presented in sufficient detail. Raw data in some cases are also difficult to interpret. For instance, there are no analyses presented of the contribution of between and among-strain variance to the total variance. The PCA approach as presented is flawed as outlined by reviewer 1. There is also a concern regarding the methodology used for QTL mapping and the lack of estimates of QTL sizes.

The recommendation for the above is to include raw and re-analyzed data along with detailed statistical analyses and discussion to address the concerns raised about the interpretation of the quantitative genetics experiments.

Reviewer 2 raises an experimental concern regarding the induction of BCAA levels by the DBT-1 cysteine variant in *C. elegans* and 293 cells that should be addressed.

Minor concerns:

The authors may also wish to take the comments from reviewer 1 into consideration regarding the scholarliness of the presentation.

Reviewer 2 raises a minor issue regarding consistency in the naming of proteins across the manuscript.

The biological system should be indicated in the title and/or Abstract:

The title and/or Abstract should provide a clear indication of the biological system under investigation (i.e., species name or broader taxonomic group, if appropriate). Please revise your title and/or Abstract with this advice in mind.

Separate reviews (please respond to each point):

*Reviewer #1:*

This paper provides an unusually broad and comprehensive approach of moving from natural variation for arsenic response in *C. elegans* to genetic mapping to gene identification to biochemical characterization. Overall, the topic is very well introduced and motivated. In some ways, the scope is almost too large for any single reviewer to assess. For this purpose, I will therefore concentrate on the genetics side of the work and let others assess the biochemistry and health implications (which does feel a bit cloying at times).

The authors use a four-fold strategy to identify the arsenic resistance locus: QTL mapping, NIL construction, GWAS, and genetic transformation via CRISPR. There can be little doubt that they have identified a gene that generates increased arsenic resistance the CB4856 line, as the allele-specific replacements via CRISPR would be the gold standard for this. On this basis, the fundamental results and conclusions of the paper appear to be on a solid foundation, providing an appropriate context for the biochemical characterization that forms the second half of the paper.

The primary concerns that I have with the work relate to the lead in to mapping results, especially around the GWAS results and how they are portrayed. While the paper definitely appears to identify "a gene" for arsenic resistance, we are given very little context for evaluating how important this gene is in the context of genetic variation for arsenic resistance per se. Instead, the mapping results are used to move directly to proposed significance in human health, but we do not really know what the relevance to the worm themselves are in the first place.

This paper clearly approaches this problem from a quantitative genetic perspective, and yet we are provided very little quantitative genetic context in which to evaluate the mapping approach, especially for the GWAS. Specifically, there is no presentation of a partitioning of within and among line variance, which is necessary to understand what kind of expectations one might have in terms of mapping outcomes, especially with respect to power. In the "transparent reporting" document that accompanies the manuscript, it is stated that "Replicates and power were determined by the throughput required for the particular assay or variance explained." First, it is not even clear what this statement means other than waving off the reporting standard being imposed by *eLife*. More importantly, however, this simply cannot be the case for the GWAS-or at least it cannot be evaluated in the absence of some sort of analysis of the quantitative genetics of the traits involved.

To attempt to address this concern, at a fairly high degree of difficulty, I worked to figure out how the data for this experiment actually worked. While we are at a brave new world of open data, which is a good thing, there could be some debate as to whether the data standard should be to simply make all raw data available or if there should be some effort to curate the data such that an external reader is told what the columns mean (e.g., what is the role of stage in the statistical outcomes), which are critical for evaluation of the results (e.g., temporal blocking), etc. I note that the Materials and methods section on "statistical analysis" outlines the most trivial model possible, so I assume that is the approach used throughout the paper. In the absence of this information, I simply tried to use a GLM analysis of the GWAS population samples to estimate (via ML) the within and among line variances across environmental treatments. I found that the among-strain variance accounted for about 1-2% of the total variance observed among individuals, while the condition-by-strain variance explained about 2-10%. Now this is for tens of thousands of observations, so these estimates are for the most part quite significant. However, it is the total among-strain (or strain-by-condition) variance that will determine the power for identifying any particular marker as significant as opposed to the overall heritability. I have little doubt that my quick and dirty look at this is flawed. However, I would expect to see something like this in the paper itself, at least as a supplementary table. Lack of this analysis also does not allow the authors to say anything about effect sizes, which are also notably absent from the paper.

It is perhaps then not surprising that no results turned up for the single-trait analysis. The authors therefore turn to a PCA approach, which itself features some fairly unusual aspects in terms of presentation. There appear to be five traits measured on each individual, yet the PCA is based on 67 measurements ("traits"). These are apparently largely summary statistics for individuals from each strain. The PCA must therefore have been generated via the among-line statistics in the absence of within-line variation, which is fine. What seems less fine is that the "traits" used here are actually all derivatives of the original five traits and are mostly descriptions of the within-line distributions. This means that many of the measures are functions of one another (e.g., the mean and CV). Indeed, I am surprised that the PCA works at all, as the correlation matrix for this set of traits should be singular. As importantly, the new PCA "traits" that emerge from such an analysis essentially have no biological meaning. They are almost entirely statistical constructions. I cannot think of any valid reason to do this, unless variance per se is the trait of interest, in which case that should be directly addressed in a much different fashion. Unless motivated quite differently, the PCA should be conducted on just the original five traits.

It is also odd that the interpretation for the PCA results is then motivated by a regression of PCA1 on body length (Supplementary Figure 4). That is exactly what the PCA loading (eigenvector coefficient) provides in the PCA analysis. If the PCA is to be conducted, then it should be done so in the context of a full analysis, which would include reporting the factor loadings. It seems that nearly all of these features simply measure how big something is (e.g., color brightness) and that including each of these measures into a single composite parameter allows the GWAS to cross a variance threshold that provides the single significant point. The PCA itself does not completely make sense from a pure statistical point of view, and so it appears that this mapping result is the primary motivation for the approach.

A final mapping oddity is that the authors report that they use full DNA strain-specific sequencing results to calculate the relatedness matrix, but only use much older RAD marker data to do the actual mapping. The only motivation that I can see for this is to reduce the number of comparisons so that the significance threshold does not become too high. Whatever the motivation, it is not specified by the authors, nor are potential power issues addressed in the paper itself or in the "transparent reporting" document.

Taken together, the GWAS approach appears to have the hallmarks of "p-hacking." In other words, the primary goal of the analysis is to ensure that there is at least one red dot on Figure 2, such that it supports the conclusions of the QTL mapping. I am not saying that this is in fact what the authors have done; simply that one gets that impression because of the statistical gymnastics that it seems to take to get here, coupled with a realistic assessment of what is actually possible to map for these traits in these populations. This view is further supported by the singular and assertive nature of the narrative, in which limitations and alternative explanations are not entertained. I fully appreciate that this is the nature of the style of presentation in top tier journals such as *eLife*, in which a clear narrative coupled to a specific mechanism seems to be a requirement for publication. However, *eLife*'s new data standards seem to be at odds with this cultural tradition. They need to figure out how they actually want to have this work. I do feel for the authors that they may be caught between an editorial expectation for clarity and the views of this specific reviewer. And I may be entirely wrong in this, which makes this specific form of publication cycle more awkward than a traditional feedback and response pattern around concerns such as this.

Similar concerns exist in the QTL mapping experiment, although the results appear to be clearer here. By my calculations the among strain variance for the RIAILs tends to be around 1-2% and the strain-by-condition variance tends to be 2-8%. Analysis of the genetic variance here is approach to the interpretation of the results, especially from the broader worm natural variation point of view, which I recognize is not a point of emphasis in the majority of the paper. Nevertheless, estimates of QTL effect sizes will help the reader interpret both these results and the interpretation of the population genetic and other functional explanations later in the paper.

The gem of the work is the allele-specific transformation that clearly demonstrates that DBT-1 has a significant effect on arsenic resistance. I do not know why these transformations were done only on the parental lines and not the NILs, although this work can be technically demanding. The latter transformations would be just a little more convincing in proving that this is actually the causal locus from the mapping experiments and not simply "a locus" that influences arsenic resistance, since there is plenty of residual difference among the parents in the transformants. However, the totality of the evidence definitely supports this interpretation, and this work is well beyond the standard in the field on this account.

I do not understand why there are no error bars on the selection experiments reported in Figure 5. Nor are there any statistical tests associated with these results that I can see. I found the raw data for these results impossible to interpret (Data File 23). Some quite important appears to be missing here, making this section impossible to interpret from a rigor and reproducibility point of view, much less in terms of scientific interpretation. I must be missing something here, so I apologize if it should be more obvious to me.

While I appreciate the desire to posit an adaptive significance on the identified alleles, we know that the central regions of *C. elegans* chromosomes (like the center of chromosome 2), so clear signs of global selective sweeps. It seems just, if not more, likely that this allele is simply along for the ride in a linked haplotype and does not in fact have any functional significance in natural populations. It is all well and good to suggest that more work is needed to explore the environmental circumstances that might generate selection, but it is somewhat disingenuous in this species to not discuss the potential for neutral processes in structuring this variation. As noted above, in general, caveats and alternative explanations are not a feature of this work, which tells a singular story from a very particular point of view. This does not make that view incorrect, but it does not seem to me to be the most rigorous way to present what is a truly impressive amount of work on a very difficult question.

*Reviewer #2:*

In this manuscript, the authors have characterized dbt-1, encoding the E2 subunit of the branched-chain α-keto dehydrogenases, as a new target for arsenic toxicity. In *C. elegans*, the arsenic treatment causes decreased brood size and reduced progeny length, but different wild type isolates show different sensitivity to the treatment. In particular, the CB4856 is more resistant. Using linkage mapping followed by NIL analysis and PCA-lined GWA mapping, the authors identified a single SNV correlated with the arsenic resistance that causes a cysteine to serine missense variant in the dbt-1 gene. Using CRSIPR/Cas9 genome editing, they further confirmed that this variant is responsible for the arsenic resistance. Given the role of DBT-1 in regulating mono-methyl branched chain fatty acids (mmBCFA) and the role of mmBCFA in regulating *C. elegans* development, the authors next compared mmBCFA levels between two variants with and without the arsenic treatment. They found that with the cysteine variant, the mmBCFA reduction is more sensitive to the arsenic treatment, and mmBCFA supplementation could rescue the arsenic sensitivity associated the cysteine variant. Furthermore, the authors also showed that the serine to cysteine alteration in the human DBT1 is associated with increased cell proliferation with the arsenic treatment. Together, this manuscript demonstrates a new link between DBT-1 and arsenic responses. There are one major and one minor issue that should be addressed before publication.

1) Major: The cysteine variant is associated with increased arsenic sensitivity in *C. elegans*, but with increased cell proliferation in 293 cells treated with arsenic. The authors suggest that the increased cell proliferation is due to the induction of BCAA levels. However this is not supported by the data using 293 cells. In *C. elegans*, the cysteine variant is associated with the induction of BCAA levels with the CB4856 background (Figure 4B). With the N2 background, the induction trend is observed but not significant different (Figure 4B). But the N2-DBT-1(S78) without arsenic has only two data sets, which might reduce the power of statistical analysis.

2) Minor: To avoid unnecessary confusion, the authors might consider changing sample labeling in Figure 4C to be consistent with other figure panels. N2 DBT-1 (S78) instead of N2 DBT-1 (C78S), CB4856 DBT-1 (C78) instead of CB4856 DBT-1 (S78C)

[Editors' note: the evaluation of the revised submission follows.]

The editors and two reviewers have re-assessed your work, "Natural variation in *C. elegans* arsenic toxicity is explained by differences in branched chain amino acid metabolism". The reviews follow below. As you will see, while both reviewers appreciate the scope and complexity of the analysis, neither reviewer is fully satisfied with the revisions and they continue to have major reservations. Given this state, the Reviewing Editor's assessment is that "major issues remain unresolved". If you elect to go forward with publication, this assessment will appear below the Abstract and at the top of the Decision Letter.

Reviewer 1 in particular remains unconvinced regarding the justification for the PCA. Reviewer 2 notes that it has not been demonstrated that the increased cell proliferation in HEK293 cells is due to increased BCAA levels. More details are in the reviews below.

*Reviewer #1:*

Thanks to the authors for addressing reviewer concerns. I would first like to start by apologizing for some of the phrasing that made it into the first review. My intention was to initiate a conversation with the other editor and other reviewers that would then be synthesized into an overall statement, as is my understanding of the *eLife* process. Thus the presence of comments related to editorial approach of the journal, which are irrelevant to the authors. I was trying to be very clear in my concerns and alternative interpretations for the benefit of the editorial process. So while the concerns that I raised are valid (and in many cases remain), my phrasing was more blunt that I would want to normally convey, especially given the scope and complexity of the work. It is my responsibility for not recognizing how the process was proceeding and not further editing my comments as part of the final decision process. This is obviously especially important given the public nature of this new review process.

With that preamble, I appreciate that the authors have clarified a great deal of the analysis, particularly the use of line means in the quantitative genetic analysis. The revision remains as the initial submission: a comprehensive analysis of natural variation in arsenic sensitivity. The overall conclusions are still well supported. Nevertheless, I find that the multivariate approach for the mapping analysis is not well justified. This need not be blown out of proportion, but must still be pointed out.

1) The authors have now greatly clarified the conversion of the raw observations into the mapping approach. This was a bit opaque before, but it is now clear that this is a line-means analysis, which is perfectly appropriate for mapping and there newly reported results show that there is substantial among-line variation (H2 for their traits). My original point relative to inferences within natural populations remains, however. These line-based differences actually explain very little about the susceptibility of individual worms to arsenic, which is largely due to random effects. The experimental design concerns discussed here mean that the value that I calculated are actually upper bound to the actual broad sense heritability in the population. This may or may not be important for the problem at hand, but it is important for the interpretation of these results from the standpoint of worm populations.

2) In the first version of the paper, the authors reported that they did not identify any significant peaks for the single trait analysis so they performed the mapping on the PCA. This is what was meant by "no results turned up for the single-trait analysis". The reviewer did not miss the summaries and supplementary materials. It did not mean that no results were presented, which is something quite different. The current manuscript has been modified to simply gloss over these issues and go directly to the PCA. While the authors "respectfully disagree" with criticisms of the PCA approach, they do not address the issues. The approach is statistically, mathematically, and biologically invalid. There is simply no justification for throwing summary statistics that are functions of one another into a PCA except to try to minimize variance. Showing that an arbitrary and invalid PCA is correlated with a biological measure begs the question as why one would not simply use the measure itself. Yes, it is possible to perform a PCA on a singular covariance matrix, but the outcomes tend to be highly unstable in the face of sampling variance. It is the eigenvectors that are the elements that are subject to interpretation. It is still the case that nothing about this analysis makes sense except that it yields a result that is consistent with the other, more convincing parts of the paper. It still feels that the main goal here was to derive a trait that yielded a significant result. The non-significant pileups of the other traits could have been subjected to joint probability test to achieve a similar goal in a much more straightforward and interpretable fashion.

3) The genomic approach and markers used in the GWAS is now much clearer.

4) The selection experiments in the human cell lines now contain error bars and significance tests, which is obviously a good thing. These effects are obvious not particularly large and selection among human cells seems like it could be subject to a number of complicating interpretations, but the result is well consistent with the overall results and interpretation.

*Reviewer #2:*

In the revised manuscript, the authors have addressed my concern on BCAA changes in *C. elegans* with more biological replicates and stronger statistical power. However, my comments on BCAA changes in 293 cells have not been addressed. To demonstrate the evolutionary conservation, the authors showed that the cysteine variant is associated with increased cell proliferation in 293 cells treated with arsenic. They suggest that this increased cell proliferation is due to the induction of BCAA levels. However this is not supported by the data using 293 cells.

---

## [Author Response]

Reviewer #1:

This paper provides an unusually broad and comprehensive approach of moving from natural variation for arsenic response in *C. elegans* to genetic mapping to gene identification to biochemical characterization. Overall, the topic is very well introduced and motivated. In some ways, the scope is almost too large for any single reviewer to assess. For this purpose, I will therefore concentrate on the genetics side of the work and let others assess the biochemistry and health implications (which does feel a bit cloying at times).The authors use a four-fold strategy to identify the arsenic resistance locus: QTL mapping, NIL construction, GWAS, and genetic transformation via CRISPR. There can be little doubt that they have identified a gene that generates increased arsenic resistance the CB4856 line, as the allele-specific replacements via CRISPR would be the gold standard for this. On this basis, the fundamental results and conclusions of the paper appear to be on a solid foundation, providing an appropriate context for the biochemical characterization that forms the second half of the paper.

We thank the reviewer for noticing the comprehensive analyses we provided and are happy to hear that the “fundamental results and conclusions of the paper appear to be on a solid foundation.”

The primary concerns that I have with the work relate to the lead in to mapping results, especially around the GWAS results and how they are portrayed. While the paper definitely appears to identify "a gene" for arsenic resistance, we are given very little context for evaluating how important this gene is in the context of genetic variation for arsenic resistance per se. Instead, the mapping results are used to move directly to proposed significance in human health, but we do not really know what the relevance to the worm themselves are in the first place.

We thank the reviewer for these observations. We addressed this criticism by quantifying and more thoroughly reporting the effect sizes of the traits that we discussed. Specifically, we included the following:

1) Effect-size calculations from the dose-response experiment (Figure 1—figure supplement 3; Figure 1—source data 3).

2) Heritability estimates from the dose-response experiment (Figure 1—figure supplement 3; Figure 1—source data 2).

3) Heritability estimates for all traits measured in the RIAIL population (Figure 1—figure supplement 7; Figure 1—source data 12)

4) Heritability estimates for all traits measured in the wild strain populations (Figure 2—figure supplement 2; Figure 2—source data 6)

5) Explicit mentions throughout the text of the effect sizes for all experiments following the mapping experiments, which include the NIL, allele-swap, and C15ISO rescue experiments. In all cases we relate our findings in these experiments to the results originally obtained in the linkage mapping experiment because the strains we used for follow-up experiments were N2 and CB4856, which were the parental strains used to generate the RIAIL population.

This paper clearly approaches this problem from a quantitative genetic perspective, and yet we are provided very little quantitative genetic context in which to evaluate the mapping approach, especially for the GWAS. Specifically, there is no presentation of a partitioning of within and among line variance, which is necessary to understand what kind of expectations one might have in terms of mapping outcomes, especially with respect to power. In the "transparent reporting" document that accompanies the manuscript, it is stated that "Replicates and power were determined by the throughput required for the particular assay or variance explained." First, it is not even clear what this statement means other than waving off the reporting standard being imposed by eLife. More importantly, however, this simply cannot be the case for the GWAS-or at least it cannot be evaluated in the absence of some sort of analysis of the quantitative genetics of the traits involved.To attempt to address this concern, at a fairly high degree of difficulty, I worked to figure out how the data for this experiment actually worked. While we are at a brave new world of open data, which is a good thing, there could be some debate as to whether the data standard should be to simply make all raw data available or if there should be some effort to curate the data such that an external reader is told what the columns mean (e.g., what is the role of stage in the statistical outcomes), which are critical for evaluation of the results (e.g., temporal blocking), etc.

We sincerely apologize for the perceived failure to report the data formats used throughout the manuscript. To address this issue, we included detailed descriptions of all columns for all Supplementary Files used in the manuscript. This information can be found at the end of the manuscript under the section labelled “Format of Supplementary Files.”

I note that the Materials and methods section on "statistical analysis" outlines the most trivial model possible, so I assume that is the approach used throughout the paper. In the absence of this information, I simply tried to use a GLM analysis of the GWAS population samples to estimate (via ML) the within and among line variances across environmental treatments. I found that the among-strain variance accounted for about 1-2% of the total variance observed among individuals, while the condition-by-strain variance explained about 2-10%. Now this is for tens of thousands of observations, so these estimates are for the most part quite significant.

We thank the reviewer for taking the time to perform statistical analysis on the GWAS data set. To confirm these results, we performed the same analysis using the raw BIOSORT data and reached the same conclusion as the reviewer. Unfortunately, these data are not what is used in our analyses. In the reviewer’s analysis, one would have to consider each observation (animal) from a population of growing animals in one well of a 96-well plate to be an independent replicate. In that well, the animals have experienced the same preparation procedure, the same bacterial food concentration, and the same environment. For this reason, we use summary statistics for each replicate population of animals grown in the same well and not single animal measurements. Our linkage and GWA mapping experiments only used a single replicate population of animals from each RIAIL or wild isolate, respectively. To make our procedure more clear, we included a “Format of Supplementary Files” section at the end of the manuscript that contains detailed information of each Supplementary File. Additionally, we added more detail to the “Calculation of fitness traits for genetic mappings” section to better describe what traits were used for the mapping and follow-up experiments.

However, it is the total among-strain (or strain-by-condition) variance that will determine the power for identifying any particular marker as significant as opposed to the overall heritability. I have little doubt that my quick and dirty look at this is flawed. However, I would expect to see something like this in the paper itself, at least as a supplementary table. Lack of this analysis also does not allow the authors to say anything about effect sizes, which are also notably absent from the paper.

As mentioned above, we have included effect-size calculations (Figure 1—source data 3) and broad-sense heritability estimates (Figure 1—source data 2) from the dose response experiment as supplemental data files. We have also reported effect sizes for the chromosome II QTL, for the NILto recapitulate the QTL effect and interval, for the allele-swap to define causality of *dbt-1*, and for the fatty acid supplementation experiments to define more of the mechanism for the effect of arsenic on *dbt-1* variation.

It is perhaps then not surprising that no results turned up for the single-trait analysis.

We are sorry that the reviewer missed our summaries and other supplemental data. Our goal was to keep the analyzed traits (brood size [norm.n] and animal length [median.TOF]) consistent throughout all experiments in the manuscript. To address this comment, we included GWA mapping results for all traits with significant QTL (Figure 2—source data 7) and included a summary plot of these results (Figure 2—figure supplement 3).These results show that non-PCA-transformed traits also map to the center of chromosome II and overlap with the QTL identified by linkage mapping. To emphasize this point, it is not just principal component traits that mapped to this QTL; 26 traits from our high-throughput fitness assays mapped to the same position. Our desire to make the traits more understandable led this reviewer down to the unfortunate conclusion that the population-wide differences in arsenic were not significant.

The authors therefore turn to a PCA approach, which itself features some fairly unusual aspects in terms of presentation. There appear to be five traits measured on each individual, yet the PCA is based on 67 measurements ("traits"). These are apparently largely summary statistics for individuals from each strain.

This assessment is correct. Please refer to our response above for further details regarding our rationale for this approach.

The PCA must therefore have been generated via the among-line statistics in the absence of within-line variation, which is fine.

We want to reiterate that we have no measure of within-line variation for the GWA mapping dataset. As we describe above, we have made the Materials and methods section more clear in this regard.

What seems less fine is that the "traits" used here are actually all derivatives of the original five traits and are mostly descriptions of the within-line distributions. This means that many of the measures are functions of one another (e.g., the mean and CV). Indeed, I am surprised that the PCA works at all, as the correlation matrix for this set of traits should be singular.

It is true that the traits we used for PCA are linearly codependent, and the determinant of the trait correlation is essentially zero (1.21e-126). However, because PCA is not iterative, in that only one eigen-decomposition step is performed, it does not require independent variables (traits) (*e.g.* a singular trait correlation matrix results in eigenvalues that approach zero). We therefore used PCA to eliminate the collinearity of the summary statistic traits. As expected from the colinearity of our traits, the PC loadings for the first principal component are very similar to each other. To further supplement this analysis, we included supplementary figures and files of the trait correlations and PC loadings for all experiments.

As importantly, the new PCA "traits" that emerge from such an analysis essentially have no biological meaning. They are almost entirely statistical constructions. I cannot think of any valid reason to do this, unless variance per se is the trait of interest, in which case that should be directly addressed in a much different fashion.

Respectfully, we disagree with this assessment. The “first principal component trait” (PC1) is a linear combination of the input traits. In the original manuscript, we showed that PC1 is highly correlated with traits associated with animal length (Figure 1—figure supplement 4-5), which is a proxy of developmental progression in our fitness assay. In an attempt to further assign biological meaning to the PC1 trait, we included PC1 vs animal length or brood size correlation plots and loadings for all experiments (Figure 1—figure supplements 2,4,10, and 11; Figure 2—figure supplement 1; Figure 3—figure supplement 2; Figure 4—figure supplement 6-7) in the revised manuscript.

Unless motivated quite differently, the PCA should be conducted on just the original five traits.

We have included additional justification for the use of using PCA throughout the manuscript. Furthermore, we relate the first principal component trait to phenotypes measured using the high-throughput fitness assay.

It is also odd that the interpretation for the PCA results is then motivated by a regression of PCA1 on body length (Supplementary Figure 4). That is exactly what the PCA loading (eigenvector coefficient) provides in the PCA analysis.

We agree with the reviewer that principal component loadings describe the same thing as what was shown in Supplemental Figure 4 of the original manuscript. However, our goal was to provide general audience *eLife* readers with a more intuitive representation of the correlation between principal components and animal length or brood size traits. Because not every reader will be as keenly aware of the intricacies of PCA as the reviewer, we think that this visualization is helpful. As stated above, we report the PC loadings for all experiments in the updated manuscript.

If the PCA is to be conducted, then it should be done so in the context of a full analysis, which would include reporting the factor loadings.

We have included all of the requested files (Figure 1—source data 4-6, 10, and 15; Figure 2—source data 2, Figure 4—source data 7) and corresponding figures (Figure 1B—figure supplements 2,4,10, and 11; Figure 2—figure supplement 1; Figure 3—figure supplement 2; Figure 4—figure supplement 6-7).

It seems that nearly all of these features simply measure how big something is (e.g., color brightness) and that including each of these measures into a single composite parameter allows the GWAS to cross a variance threshold that provides the single significant point. The PCA itself does not completely make sense from a pure statistical point of view, and so it appears that this mapping result is the primary motivation for the approach.

We hope that the reviewer understands the approach better after the clarifications above and our edits to the manuscript.

A final mapping oddity is that the authors report that they use full DNA strain-specific sequencing results to calculate the relatedness matrix, but only use much older RAD marker data to do the actual mapping.

We appreciate this criticism. The use of the RAD marker data since its original publication in 2012 has been to keep markers consistent among different *C. elegans* GWA mapping experiments. We agree that this approach makes less sense with genome-wide sequence data available. We found (though we did not discuss in the original manuscript) that these markers are sufficient to “tag” all strain haplotypes present in 97 wild isolates made available by Andersen et al., 2012. The 86 strains that we phenotyped in the presence of arsenic trioxide were drawn from this same 97 strain population. We reasoned that the same marker set would be sufficient for GWA mapping. However, we updated our GWA mapping approach to alleviate any possible concerns the reviewer might have about our choice of markers. Here is a brief summary that we detail in the Materials and methods section:

We started with the imputed VCF that can be found at https://elegansvariation.org/data/release/latest. We removed variants with minor allele frequency less than 5% and then removed highly correlated markers ((r^2) > 0.8). Finally, we kept sites where all strain genotypes were present. This pruned marker set contains 59,241 markers. We used this marker set to construct the relatedness matrix and perform GWA mapping on the 86 phenotyped strains. We have included the resulting marker set in Figure 2—source data 5.

The only motivation that I can see for this is to reduce the number of comparisons so that the significance threshold does not become too high. Whatever the motivation, it is not specified by the authors, nor are potential power issues addressed in the paper itself or in the "transparent reporting" document.

We are sorry that the reviewer evaluated our analysis and came to this conclusion. As we described, a large number of traits map to the same QTL on chromosome II above a Bonferroni-corrected significance threshold. The updated manuscript has several changes. To improve our mapping techniques, we included genome-wide markers and then used two different methods for setting significance thresholds. The first is the Bonferroni-corrected (BF) threshold as we already described. The BF threshold for the updated marker set is ~ 8.4e-07, corresponding a -log10(p) of ~ 6. In the second method for multiple-testing correction, we performed spectral decomposition of the genotype correlation matrix from the genome-wide marker data set as performed previously. We set all eigenvalues greater than one to one and then summed the all of the eigenvalues to identify the number of independent tests in the marker set. This analysis revealed that there are 500.761 independent tests in our marker set of 59,241 markers, corresponding to a significance threshold of 1e-04 or a -log10(p) of 4. We have included a detailed explanation of this approach in the Materials and methods section.

Taken together, the GWAS approach appears to have the hallmarks of "p-hacking." In other words, the primary goal of the analysis is to ensure that there is at least one red dot on Figure 2, such that it supports the conclusions of the QTL mapping.

We are convinced that the main drivers of the reviewer reaching this conclusion are (1) a misunderstanding of the input data structure and (2) marker set choice. We have addressed both of these points above. However, the reviewer’s suspicion motivated us to update our mapping approach, and we are thankful for this input.

The result of our updated GWA mapping pipeline resulted in 26 of the 64 traits that we used as input for PCA to map to the center of chromosome II. As for the PC1 trait, the p-value for the peak marker of the chromosome II QTL is ~1.48e-07, which is nearly an order of magnitude greater than the Bonferroni-corrected significance threshold and three orders of magnitude greater than the spectral decomposition threshold described above. If we were to take multiple testing correction to the extreme and set the number of tests to be 64 traits tested * 60,000 markers we end up with ~ 4 million tests (BF threshold: ~1.3e-8) and 33,000 tests (decomposition threshold: ~1.5e-6). While the PC1 QTL peak p-value does not quite pass the BF significance threshold in this extreme case, it does pass the decomposition threshold by an order of magnitude. We note that some of the other traits do pass this extreme threshold (mean.yellow: p-value at peak marker ~ 2.6e-10) However, as the reviewer pointed out, we do not have 64 independent traits because they are all summary statistics of five traits quantified by the BIOSORT. This observation is supported by our observation that four principal components capture >90% of the total variance in the GWAS phenotype data set. If we were to perform multiple testing correction across these four traits, which we note are still not fully independent, the BF threshold is ~2.1e-07, which is nearly two times higher than the p-value for the chromosome II QTL peak marker obtained from mapping the PC1 trait, making our mappings significant.

I am not saying that this is in fact what the authors have done; simply that one gets that impression because of the statistical gymnastics that it seems to take to get here, coupled with a realistic assessment of what is actually possible to map for these traits in these populations.

Again, we are sorry that the reviewer reached this conclusion. We hope our explanations and re-analyses are sufficiently convincing.

This view is further supported by the singular and assertive nature of the narrative, in which limitations and alternative explanations are not entertained. I fully appreciate that this is the nature of the style of presentation in top tier journals such as eLife, in which a clear narrative coupled to a specific mechanism seems to be a requirement for publication. However, eLife's new data standards seem to be at odds with this cultural tradition. They need to figure out how they actually want to have this work. I do feel for the authors that they may be caught between an editorial expectation for clarity and the views of this specific reviewer. And I may be entirely wrong in this, which makes this specific form of publication cycle more awkward than a traditional feedback and response pattern around concerns such as this.

Our main goal was to describe the results of our analyses that clearly show variation in the *dbt-1* locus contributes to differential arsenic toxicity in the *C. elegans* population. We have provided all raw and processed data, included a description of data processing and mapping procedures, and expanded our reported traits to emphasize that this QTL is highly significant. The reviewer’s comments have improved the overall clarity of the manuscript and results. We are grateful for the input.

Similar concerns exist in the QTL mapping experiment, although the results appear to be clearer here. By my calculations the among strain variance for the RIAILs tends to be around 1-2% and the strain-by-condition variance tends to be 2-8%.

The phenotyping of the RIAILs used for linkage mapping was performed in the same manner as we described above for the wild isolates used for GWA mapping. Specifically, we did not quantify replicates of the RIAIL phenotypes, instead we relied on the replication of RIAIL genotypes being derived from two parental strains. This precludes proper analysis of variance for this particular data set, but we are able to estimate variance components based on genetic relatedness across the genome, as we describe below.

Analysis of the genetic variance here is approach to the interpretation of the results, especially from the broader worm natural variation point of view, which I recognize is not a point of emphasis in the majority of the paper. Nevertheless, estimates of QTL effect sizes will help the reader interpret both these results and the interpretation of the population genetic and other functional explanations later in the paper.

It is true that our main emphasis was not focused on a more quantitative genetic analysis, including genetic variance in our dataset. However, to address the reviewers concerns thoroughly, we added the following analyses:

1) We have provided effect size estimates for the dose response experiment for all measured and principal component traits. We report different measures of effect size including η2, partial η2, ω2, partial ω2, and Cohen’s F. Additionally, we included broad-sense heritability estimates from this experiment. – Supplementary Files 2-3 and Figure 1—figure supplement 3.

2) We have included genomic heritability estimates of all BIOSORT and principal component traits used for linkage mapping. We provide two estimates of narrow- and broad-sense heritability using a linear mixed-effect model with the equation y=Xβ+Zu+ϵ. The estimates differ in their formulation for the strain additive relatedness matrix, realized relatedness matrices correct for allele frequencies and the expectation matrix does not. For both estimates, the Hadamard product of the additive relatedness matrix was used to calculate the epistatic relatedness matrix (Figure 1—figure supplement 7; Figure 1—source data 12).

3) We have provided effect size estimates for the QTL identified by linkage mapping (Figure 1—source data 11), including discussion of how much of the total additive variance is explained by the identified QTL.

4) We have included presentation of the QTL effect size identified by GWAS in the manuscript, as well as discussion of variance partitioning (Figure 2—figure supplement 2; Figure 2—source data 6).

5) We also present how much of the parental phenotypic difference can be explained by the NILs and genome-edited allele-replacement strains.

The gem of the work is the allele-specific transformation that clearly demonstrates that DBT-1 has a significant effect on arsenic resistance. I do not know why these transformations were done only on the parental lines and not the NILs, although this work can be technically demanding. The latter transformations would be just a little more convincing in proving that this is actually the causal locus from the mapping experiments and not simply "a locus" that influences arsenic resistance, since there is plenty of residual difference among the parents in the transformants. However, the totality of the evidence definitely supports this interpretation, and this work is well beyond the standard in the field on this account.

We agree with the reviewer that the allele-specific replacements clearly demonstrate that DBT-1 variation has a significant effect on arsenic resistance. We did not find it necessary to introduce the allele-replacements in the NIL genetic background because these genome-edited strains completely recapitulated the NIL effect. However, we realize that, because we did not include more discussion of effect size recapitulation by the NIL and allele-swap strains, it may not be immediately obvious that the DBT-1(C78S) allele completely recapitulated the QTL effect identified via linkage mapping. Specifically, the NILs and allele-swap strains account for ~64-92% of the parental difference in the presence of arsenic trioxide and the chromosome II QTL, which explains ~33% of the phenotypic variation in the RIAIL population, accounts for ~61.7% of the total phenotypic variance explained by genetic factors (H2~ 0.53 for PC1). The discrepancy between the mapping and follow-up experiments is within error of the heritability estimate. We included discussion of these points in the revised manuscript.

I do not understand why there are no error bars on the selection experiments reported in Figure 5. Nor are there any statistical tests associated with these results that I can see. I found the raw data for these results impossible to interpret (Data File 23). Some quite important appears to be missing here, making this section impossible to interpret from a rigor and reproducibility point of view, much less in terms of scientific interpretation. I must be missing something here, so I apologize if it should be more obvious to me.

We updated this figure to include error bars, and we calculated significance associated with differences in read counts across all replicates using Fisher’s exact test (Figure 5—source data 2). We also updated the description of the data file corresponding to these read data to make it more easily interpretable for future readers.

While I appreciate the desire to posit an adaptive significance on the identified alleles, we know that the central regions of *C. elegans* chromosomes (like the center of chromosome 2), so clear signs of global selective sweeps.

It is our understanding that chromosomes I, IV, V, and X have undergone selective sweeps, not chromosome II. Nevertheless, we explicitly state that the genomic region surrounding *dbt-1* has no signature of selection as indicated by Tajima’s D, and we did not explore this possibility further.

It seems just, if not more, likely that this allele is simply along for the ride in a linked haplotype and does not in fact have any functional significance in natural populations. It is all well and good to suggest that more work is needed to explore the environmental circumstances that might generate selection, but it is somewhat disingenuous in this species to not discuss the potential for neutral processes in structuring this variation.

We have added to the Discussion section to address the possibility of no adaptive advantage associated with this allele.

As noted above, in general, caveats and alternative explanations are not a feature of this work, which tells a singular story from a very particular point of view. This does not make that view incorrect, but it does not seem to me to be the most rigorous way to present what is a truly impressive amount of work on a very difficult question.

We added alternative explanations throughout the manuscript as described in our responses to the previous comments from this reviewer. We hope that these additions have shown the breadth of our analyses that led to an interesting discovery.

Reviewer #2:

[…] There are one major and one minor issue that should be addressed before publication.1) Major: The cysteine variant is associated with increased arsenic sensitivity in *C. elegans*, but with increased cell proliferation in 293 cells treated with arsenic. The authors suggest that the increased cell proliferation is due to the induction of BCAA levels. However this is not supported by the data using 293 cells. In *C. elegans*, the cysteine variant is associated with the induction of BCAA levels with the CB4856 background (Figure 4B). With the N2 background, the induction trend is observed but not significant different (Figure 4B). But the N2-DBT-1(S78) without arsenic has only two data sets, which might reduce the power of statistical analysis.

We thank the reviewer for this observation. We thought that, in the context of the CB4856 and CB4856 allele swap result, the conclusion was solid as is. However, we performed the L1 larval stage metabolite profiling experiment at higher replication to address the reviewers’ point. For this experiment, we only included the N2 and N2 allele swap strains because these are the relevant strains to make the conclusion that there is a DBT-1 allele-specific effect on branched-chain fatty acid production. We acquired an additional six independent paired 100 µM arsenic trioxide-control replicates to strengthen our conclusions. The results of this experiment are described in the “Arsenic trioxide inhibits the DBT-1 C78 allele” section of the manuscript and shown in Figure 4B and Figure 4—figure supplements 2-4. The results from this experiment support the conclusion that strains with the DBT-1(C78) have higher branched/straight-chain ratios relative to strains with the DBT-1(S78) for the C17 (CB4856 DBT-1(C78): Tukey HSD *p-*value = 0.164721, n=3; N2 DBT-1(C78): Tukey HSD *p-*value = 0.003747, n=6) and C15 ratios (CB4856: Tukey HSD *p-*value = 0.0427733, n=3; N2: Tukey HSD *p-*value = 0.0358, n=6). We note that the C17ISO/C17SC ratio is not significantly different when comparing the CB4856 and CB4856 allele swap strain, but the direction of effect matches our other observations. Furthermore, we show that there are DBT-1 allele-specific differences in C15ISO and C17ISO for all strains in control conditions (CB4856-C15ISO DBT-1(C78): Tukey HSD *p-*value = 0.0036201, n=3; N2-C15ISO DBT-1(C78): Tukey HSD *p-*value = 0.0265059, n=6; CB4856-C17ISO DBT-1(C78): Tukey HSD *p-*value = 0.0086572, n=3; N2-C17ISO DBT-1(C78): Tukey HSD *p-*value = 0.0022501, n=6).Importantly, the DBT-1 allele-specific differences in the fatty acid ratio and ISO measurements were not driven by differences in straight-chain fatty acids.

In addition to increasing the replication of the L1 assay, we quantified the effects of the DBT-1(C78S) allele in young adult animals to see if the conclusions held true across different developmental stages. For this experiment, we included six independent replicates for the N2, CB4856, and allele-swap strains. In contrast to the results at the L1 developmental stage, we did not observe the same effect of the DBT-1(C78S) allele at the young adult life stage. The results of this experiment are shown in Figure 4—figure supplements 5.

Taken together, these results suggest that the DBT-1(C78) allele produces more branched chain fatty acids than the DBT-1(S78) allele, but this effect is dependent on the developmental stage of the animals.

2) Minor: To avoid unnecessary confusion, the authors might consider changing sample labeling in Figure 4C to be consistent with other figure panels. N2 DBT-1 (S78) instead of N2 DBT-1 (C78S), CB4856 DBT-1 (C78) instead of CB4856 DBT-1 (S78C).

We updated Figure 4C to make the genotypes clearer.

[Editors' note: the evaluation of the revised submission follows.]

Reviewer 1 in particular remains unconvinced regarding the justification for the PCA. Reviewer 2 notes that it has not been demonstrated that the increased cell proliferation in HEK293 cells is due to increased BCAA levels. More details are in the reviews below.Reviewer #1:[…] With that preamble, I appreciate that the authors have clarified a great deal of the analysis, particularly the use of line means in the quantitative genetic analysis. The revision remains as the initial submission: a comprehensive analysis of natural variation in arsenic sensitivity. The overall conclusions are still well supported. Nevertheless, I find that the multivariate approach for the mapping analysis is not well justified. This need not be blown out of proportion, but must still be pointed out.1) The authors have now greatly clarified the conversion of the raw observations into the mapping approach. This was a bit opaque before, but it is now clear that this is a line-means analysis, which is perfectly appropriate for mapping and there newly reported results show that there is substantial among-line variation (H2 for their traits). My original point relative to inferences within natural populations remains, however. These line-based differences actually explain very little about the susceptibility of individual worms to arsenic, which is largely due to random effects. The experimental design concerns discussed here mean that the value that I calculated are actually upper bound to the actual broad sense heritability in the population. This may or may not be important for the problem at hand, but it is important for the interpretation of these results from the standpoint of worm populations.

Thank you for your comment about the revision. We added substantial explanations, additional data, and a data dictionary to explain our approach. It is a line-means analysis. Because *C. elegans* is a selfing hermaphrodite with little diversity at the local level, most populations are nearly clonal. Our approach does not address the influences on individual animals but does address what a nearly genetically identical population would encounter in the wild. We added some details to the Results about how this analysis might not be applied easily to natural populations at the individual level.

2) In the first version of the paper, the authors reported that they did not identify any significant peaks for the single trait analysis so they performed the mapping on the PCA. This is what was meant by "no results turned up for the single-trait analysis". The reviewer did not miss the summaries and supplementary materials. It did not mean that no results were presented, which is something quite different. The current manuscript has been modified to simply gloss over these issues and go directly to the PCA. While the authors "respectfully disagree" with criticisms of the PCA approach, they do not address the issues. The approach is statistically, mathematically, and biologically invalid. There is simply no justification for throwing summary statistics that are functions of one another into a PCA except to try to minimize variance. Showing that an arbitrary and invalid PCA is correlated with a biological measure begs the question as why one would not simply use the measure itself. Yes, it is possible to perform a PCA on a singular covariance matrix, but the outcomes tend to be highly unstable in the face of sampling variance. It is the eigenvectors that are the elements that are subject to interpretation. It is still the case that nothing about this analysis makes sense except that it yields a result that is consistent with the other, more convincing parts of the paper. It still feels that the main goal here was to derive a trait that yielded a significant result. The non-significant pileups of the other traits could have been subjected to joint probability test to achieve a similar goal in a much more straightforward and interpretable fashion.

We are sorry about this issue. As the reviewer points out, the PCA approach was used in the first version of the paper to reduce the noise from the 67 individual trait measures. As we stated before, the approach was not used to yield a result “consistent with the other, more convincing parts of the paper”. The original genome-wide association mappings in the first version of the manuscript used the RAD-sequencing marker set and a Bonferroni-corrected p-value threshold on only two traits (brood size and median animal length). For these two traits, we did not map a QTL to the center of chromosome II, which we hypothesized was caused by noisy traits measured from single replicates of a small number of wild strains. However, other traits did map to this chromosome II region as we presented in the resubmission. For this reason, we used the PCA approach to “clean” up the correlated traits and performed the mappings to find a QTL on chromosome II. It was not performed after the linkage mappings, NIL experiments, and allele-replacement tests. As we described previously, the overlap of these two mapping approaches is what inspired us to perform the follow-up experiments. In the revision (and thanks to the reviewer’s comments), we updated our mapping algorithm, the marker data set, added a new significance threshold, and included the mapping results of all of the traits used as inputs to PCA. These changes enabled significant GWA mappings (Bonferroni correction) for 26 of 64 of the summary BIOSORT-measured traits and the major PC to the same position on the center of chromosome II (Author response image 1). All trait mapping results from the original revision can be found on FigShare (https://doi.org/10.6084/m9.figshare.7458932.v2). In our revision, we attempted to justify the use of PCA again as a data cleaning method to show that the correlated summary statistics were loaded into the first PC. After the reviewer’s latest comments and discussions with mathematicians, we understand the limitations of using PCA to “clean” up correlated summary statistics as opposed to looking for relationships among seemingly unrelated traits. We would like to point out again that, in our original revision, 26 of 64 statistical summary traits, which relate directly to the biology of *C. elegans (e.g.* the 90th quantile of animal length), map significantly (Author response image 1) to the same position on chromosome II. To follow the reviewer’s suggestion and the editor’s request for a more appropriate PCA approach, we only used four traits (brood size: norm.n; animal length: mean.TOF; optical density: mean.norm.EXT; and fluorescence: mean.norm.yellow), which are each measured independently from each other (*i.e.* different lasers/PMTs or counts on the COPAS BIOSORT), as inputs for PCA. As mentioned above, these four traits are correlated, so it is clear that PC1 is capturing overall arsenic-induced toxicity, and this PC mapped to the center of chromosome II. We use this PC trait across all figures to keep consistency across all analyses. It is important to note that this PC is also highly correlated with biologically meaningful traits like population animal length and optical density. The overlap of association and linkage mapping experiments led us to the interval tested using NILs and containing the allele-replacement validated gene *dbt-1*. Of the independent traits that we used as inputs for the PCA, only the brood size trait alone did not map to the center of chromosome II. We also include the analysis and results of these four independently quantified traits for all follow-up experiments. We would like to emphasize that this new analysis does not change any of the conclusions of the manuscript, but we hope these final changes address the reviewer’s remaining concerns regarding the PCA approach.

**Author response image 1. respfig1:** GWA mapping QTL summary. All QTL identified by GWA mapping are shown. Traits are labeled on the y-axis, and the genomic position in Mb is plotted on the x-axis. Triangles represent the peak QTL position, and bars represent the associated QTL region of interest. Triangles and bars are colored based on the significance value, where red colors correspond to higher significance values.

3) The genomic approach and markers used in the GWAS is now much clearer.

Thank you. We are much happier with the new GWAS approach, which was motivated by the reviewer’s initial concerns. As we explained in the manuscript and above, the majority of traits, including the first principal component trait, had at least one significant QTL on chromosome II overlapping *dbt-1*. These results show that natural variation in arsenic responses across *C. elegans* wild strains maps by linkage mapping to the center of chromosome II (shown here as Author response image 2).

**Author response image 2. respfig2:** Linkage mapping QTL summary. All QTL identified by linkage mapping are shown. Traits are labeled on the y-axis, and the genomic position in Mb is plotted on the x-axis. Triangles represent the peak QTL position, and bars represent the associated 1.5-LOD drop QTL confidence interval. Triangles and bars are colored based on the LOD score, where red colors correspond to higher LOD values.

4) The selection experiments in the human cell lines now contain error bars and significance tests, which is obviously a good thing. These effects are obvious not particularly large and selection among human cells seems like it could be subject to a number of complicating interpretations, but the result is well consistent with the overall results and interpretation.

Thank you for these comments.

Reviewer #2:

In the revised manuscript, the authors have addressed my concern on BCAA changes in *C. elegans* with more biological replicates and stronger statistical power. However, my comments on BCAA changes in 293 cells have not been addressed. To demonstrate the evolutionary conservation, the authors showed that the cysteine variant is associated with increased cell proliferation in 293 cells treated with arsenic. They suggest that this increased cell proliferation is due to the induction of BCAA levels. However this is not supported by the data using 293 cells.

This error was ours and should have been avoided. During the editing process between different software packages, the offending sentence was not removed in the submitted revision. We have removed this sentence and also tightened the arguments in human cells (final section of the Results). We hope that these changes have alleviated this concern.